

# Quantification and characterization of primary biological aerosol particles and bacteria aerosolized from Baltic seawater

Julika Zinke[1,2,3], Gabriel Freitas[1,2], Rachel A. Foster[4], Paul Zieger[1,2], E. Douglas Nilsson[1,2], Piotr Markuszewski[1,2,5], and Matthew E. Salter[1,2,3]

[1]Department of Environmental Sciences, Stockholm University, Stockholm, Sweden
[2]Bolin Centre for Climate Research, Stockholm University, Stockholm, Sweden
[3]Now at the Baltic Sea Centre, Stockholm University, Stockholm, Sweden
[4]Department of Ecology, Environment and Plant Sciences, Stockholm University, Stockholm, Sweden
[5]Institute of Oceanology, Polish Academy of Science, Sopot, Poland

**Correspondence:** Julika Zinke (julika.zinke@su.se)

**Abstract.** Primary biological aerosol particles (PBAP) can influence climate and affect human health. To investigate the aerosolization of PBAP with sea spray aerosol (SSA), we conducted ship-based campaigns in the central Baltic Sea near Östergarnsholm in May and August 2021. Using a plunging jet sea spray simulation chamber filled with local seawater, we performed controlled chamber experiments to collect filters and measure aerosols. We determined the abundance of bacteria in the chamber air and seawater by staining and fluorescence microscopy, normalizing these values to sodium concentration to calculate enrichment factors. Our results showed that bacteria were enriched in the aerosol by 13 to 488 times compared to the underlying seawater, with no significant enrichment observed in the sea surface microlayer. Bacterial abundances obtained through microscopy were compared with estimates of fluorescent PBAP (fPBAP) using a single-particle fluorescence spectrometer. We estimated bacterial emission fluxes using two independent approaches: (1) applying the enrichment factors derived from this study with mass flux estimates from previous SSA parameterizations, and (2) using a scaling approach from a companion study. Both methods produced bacterial emission flux estimates that were in good agreement and on the same order of magnitude as previous studies, while fPBAP emission flux estimates were significantly lower. Furthermore, 16S rRNA sequencing identified the diversity of bacteria enriched in the nascent SSA compared to the underlying seawater.

## 1 Introduction

Primary biological aerosol particles (PBAP) are airborne particles that include microbes (bacteria, archaea, pico-eukaryotes), viruses, pollen, and spores, which can exist as agglomerates, single particles, or cell fractions. Before aerosolization, these particles are referred to as microorganisms. Although PBAP constitute less than $0.1\%$ of aerosol particles by number (Mayol et al., 2017), they play a significant role in the chemical and biological composition of aerosols. For instance, PBAP are efficient cloud and ice condensation nuclei (Wilson et al., 2015; DeMott et al., 2016; Šantl-Temkiv et al., 2019), potentially affecting cloud properties such as phase, albedo, and lifetime, thus influencing Earth's climate and biogeochemical cycles (Fröhlich-





Nowoisky et al., 2016). Additionally, PBAP can impact human health and well-being (Genitsaris et al., 2011; Smets et al., 2016; May et al., 2018), though their specific effects remain inadequately understood (Alsante et al., 2021).

Once aerosolized, PBAP can be transported over hundreds to thousands of kilometers, with residence times ranging from days to weeks (Mayol et al., 2017). The transport and residence time of PBAP depend on their size, density, shape (Tesson and Šantl-Temkiv, 2018), and atmospheric conditions such as wind speed, direction, and precipitation. The impact of marine PBAP on ecosystems is influenced by their fluxes between the ocean and atmosphere, their atmospheric transport distance and altitude, and their viability during transport (Alsante et al., 2021). Therefore, it is crucial to better characterize the air-sea exchange of marine microorganisms, their dispersal in the atmosphere, and their ability to adapt to atmospheric conditions.

PBAP from the oceans are released into the atmosphere along with sea spray aerosols (SSA), a significant natural aerosol source. SSA forms when waves break and trap air as bubbles in the seawater. These bubbles rise to the surface, scavenging biogenic material , and burst at the surface, generating film drops film drops from the disintegration of the bubble film cap and jet drops resulting from the collapse of the bubble cavity. Film drops are numerous, typically smaller than $< 1\,\mu m$ and are enriched in organic materials from the surface microlayer (SML) such as cell fragments, and small microorganisms such as bacteria and viral particles (Blanchard and Syzdek, 1982; Rastelli et al., 2017; Michaud et al., 2018). The SML has a distinct microbial community composition and hydrographic conditions compared to the underlying water (Franklin et al., 2005; Joux et al., 2006; Cunliffe et al., 2009; Stolle et al., 2011). In contrast, jet drops are typically larger than 1 μm in radius and consist mainly of sea salt, water-soluble organics, and larger microorganisms from underlying waters (Wang et al., 2017). Recent observations indicate that the contribution of the SML to SSA is relatively small compared to bubbles originating from subsurface water (Chingin et al., 2018; Frossard et al., 2019).

Our understanding of the atmospheric abundance and diversity of PBAP over open oceans is limited due to spatial and temporal sampling constraints. Additionally, the low concentration of PBAP necessitates either high sample flows or long sampling times to obtain sufficient biomass for downstream analyses such as microscopy, flow cytometry, or DNA sequencing. Consequently, the role of oceans as a source or sink of PBAP is not fully resolved (Burrows et al., 2009b, a; Amato et al., 2023), particularly concerning bacterial emission fluxes in remote open ocean and coastal regions (Amato et al., 2023).

Estimates of bacterial emission fluxes from the oceans vary, with global averages around $60\,\text{cells}\,\text{m}^{-2}\text{s}^{-1}$ (Burrows et al., 2009b) and regional studies showing fluxes between 10 and $100\,\text{cells}\,\text{m}^{-2}\text{s}^{-1}$ (Mayol et al., 2014, 2017; Hu et al., 2017). However, direct measurements of bacterial emission fluxes are rare due to the lack of suitable bioaerosol measurement techniques for traditional eddy correlation flux measurements. Recent advancements in real-time single-particle analysis instruments using ultra-violet light-induced fluorescence allow continuous monitoring of fluorescent (fPBAP) (e.g. Huffman et al., 2020; Santander et al., 2021; Freitas et al., 2022). Despite this progress, these instruments are limited to detecting particles larger than some microorganisms (e.g. bacteria as small as 0.2 μm in diameter, Schulz and Jørgensen, 2001) and may underestimate the total PBAP abundance. Additionally, these instruments cannot provide information on the diversity of microbial community composition, which is important because studies have shown significant variations in airborne microbial communities across different oceanic regions (Seifried et al., 2015; Michaud et al., 2018; Mayol et al., 2017; Lang-Yona et al., 2022). Comprehensive global studies on the diversity of airborne bacteria, especially in pristine marine sites, are still lacking, however.



Currently, only one study of global airborne microbial communities exists, and it does not include pristine marine sampling sites. (Tignat-Perrier et al., 2019).

Previous mesocosm studies have shown that bacteria are significantly enriched in SSA compared to underlying seawater (Carlucci and Williams, 1965; Cipriano, 1979; Blanchard and Syzdek, 1972, 1982; Marks et al., 2001; Aller et al., 2005; Rastelli et al., 2017; Zinke et al., 2024b), with enrichment factors ranging between 10 and 2500. Advancements in both culture-based and culture-independent approaches have provided new insights into airborne bacterial communities. Certain microbial taxa, particularly those with hydrophobic surface properties, have been found to be selectively aerosolized (Fahlgren et al., 2015; Rastelli et al., 2017; Perrott et al., 2017; Michaud et al., 2018; Freitas et al., 2022; Zinke et al., 2024b). These hydrophobic properties may enhance their transport to the sea surface and their inclusion in SSA. Fahlgren et al. (2015) suggested that pigmentation might also influence selective aerosolization by affecting surface properties. Marks et al. (2019) proposed that the enrichment of bacteria and diatoms in SSA could result from anionic bubble surfaces attracting cells with typically negative charges on their outer membranes. When these bubbles reach the water surface and burst, the collected microorganisms are ejected into the air with the initial or secondary jet droplets projected upward from the sub-bubble cationic vortex.

In this study, we conducted two ship-based campaigns in the Baltic Sea to investigate the air-sea exchange of fPBAP, focusing on bacteria in SSA using chamber experiments. We aimed to quantify the contribution of marine fPBAP to the atmospheric aerosol load and identify bacterial taxa preferentially aerosolized with SSA. We employed filter-based sampling for airborne bacteria and continuous fPBAP measurements, calculated bacterial enrichment factors in SSA compared to subsurface seawater, and used two approaches to estimate bacterial emission flux from coastal Baltic Sea areas. Additionally, 16S rRNA sequencing of subsurface seawater, SML, and chamber-generated SSA samples allowed us to investigate the enrichment of specific taxa in SML and aerosol relative to underlying seawater.

## 2 Methods

### 2.1 Experimental set-up and sample collection

Aerosol and seawater samples were collected during two cruises in the Baltic Sea aboard the Research Vessel (R/V) *Oceania* (18 to 29 May 2021) and R/V *Electra* (9 to 22 August 2021). Both ships were stationed near the Integrated Carbon Observation System (ICOS) eddy covariance flux station on Östergarnsholm island, east of Gotland (57°25'48.4" N, 18°59'02.9" E). In addition to the station at Östergarnsholm (between 19 and 24 May), R/V *Oceania* also conducted a transect through the Baltic proper (Fig. S1).

Nascent SSA was generated using a plunging jet sea spray simulation chamber (total volume of 200 L) filled with local seawater (volume of 90 L) from the ship's flow-through system at a depth of 1.5 m. This chamber, described in detail in Salter et al. (2014), allowed us to collect nascent SSA on filters and conduct online aerosol measurements under controlled conditions, excluding terrestrial sources, while also monitoring seawater properties. To prevent contamination from ambient air, the chamber head space was continuously flushed with particle-free air generated by a dry air generator equipped with a HEPA filter (Kaeser, model Dental T, Germany during the *Oceania* campaign, and Dürr Dental SE, model Silver Airline Trio 160



lpm, Germany, during the *Electra* campaign). The chamber can operate in flow-through mode, continuously replacing seawater
from the ship's seawater inlet. However, during the second campaign, whenever *Electra* had to leave its anchored position and
return to harbour, the chamber operated in closed mode, meaning that the seawater was recirculated in the chamber until the
ship was back on station.

During the R/V *Oceania* campaign, surface seawater salinity and temperature were measured with a thermosalinograph
(SBE21, Sea-Bird Scientific, USA) and dissolved oxygen in the water was measured with an oxygen meter (Fibox 4 trace,
PreSens Precision Sensing GmbH, Germany) in line with the flow-through system. During the R/V *Electra* campaign, seawater
salinity and temperature were measured with a conductivity sensor (Aanderaa 4120, Norway), and dissolved oxygen was
measured with an optode (Aanderaa 4175, Norway) inside the chamber. The daily average concentrations of chlorophyll-*a* in
the seawater at the location where the ships were anchored close to Östergarnsholm was determined using re-analysis data
(Woźniak et al., 2011a, b; Konik et al., 2019).

Bulk aerosol samples were collected using filters attached to the sea spray simulation chamber, positioned approximately
45 cm above the water surface. Samples for bacteria enumeration and ion chromatography were collected on black polycarbon-
ate filters (pore size 0.2 µm, 25 mm diameter, Merck Millipore Ltd., Ireland), while samples for DNA analysis were collected
on supor filters (0.2 µm pore sized, 25 mm diameter, Pall Corporation, USA). Aerosol filter samples were collected for 24
hours at a flow rate of $5\,\mathrm{L\,min^{-1}}$ controlled by mass flow controllers (hereafter referred to as MFC, model 5850E, Brooks
instruments, USA).

Water samples for bacterial enumeration (100 mL), ion chromatography (10 mL), and DNA analysis (500 mL) were collected
each time the aerosol filters were exchanged. Additionally, SML samples (500 mL total volume) for bacterial enumeration, ion
chromatography, and DNA analysis were collected once per day during the R/V *Oceania* campaign by lowering a Garett
screen (Garrett, 1965) over the side of the ship. During the R/V *Electra* campaign, SML samples (500 mL) were collected from
110 a smaller boat using a glass plate (20 x 35 cm) and a squeegee (Harvey and Burzell, 1972).

Low biomass in aerosol samples, as collected in this study, poses a risk of contamination, making it crucial to ensure steril-
ized conditions, use sterile techniques, and take operational blanks to verify that measurements are not biased by contamination
(Šantl-Temkiv et al., 2020; Dommergue et al., 2019). To ensure sterility, the sampling equipment (sampling bottles, filter hold-
ers and tubes) was autoclaved prior to each campaign. Additionally, equipment used repeatedly was cleaned in a 10% bleach
bath between each sampling time. Handling blanks were collected throughout the campaigns for each analysis method. This
involved placing the filters in their respective filter holders without drawing any air through them and immediately removing
them again. All samples were stored at -20°C until further analysis.

A total of 11 filter samples were collected each during the R/V *Oceania* and *Electra* campaigns. However, due to poor
staining, bacteria could not be confidently enumerated during the R/V *Oceania* campaign, so these samples will only be
discussed in terms of bacterial community composition. Conversely, samples collected during the R/V *Electra* campaign will
be discussed in terms of bacterial abundance, enrichment, and community composition. For more details, an overview of all
samples and the methods applied to each sample can be found in the supplement (Table S1 and Table S2).



## 2.2 Bacteria enumeration

After collection, the filter samples designated for bacteria enumeration underwent immediate sonication in a sonicator bath
(model 1510, Branson Ultrasonics, USA) at 40 Hz for one minute in 5 mL of ultrapure water to extract the bacteria from
the filters. An aliquot of 0.5 mL was reserved for ion chromatography analysis. The remaining suspension was chemically
fixed with 4% paraformaldehyde solution (w:v) for 45 minutes and stained with 4',6-diamidino-2-phenylindole (DAPI) at a
concentration of 10 μg mL$^{-1}$ for 15 minutes. The stained suspension was then filtered through another black polycarbonate
filter (0.2 μm pore size, 25 mm diameter, Merck Millipore Ltd., Ireland) rinsed with phosphate-buffered saline (1X PBS, pH
7.4). Subsequently, the filters were mounted on slides using an anti-fade solution (Epredia Lab Vision PermaFluor Aqueous
Mounting Medium, Fisher Scientific, Sweden) along with cover slips, and stored at -20 °C until enumeration was conducted
on land. Similarly, handling blank filters were processed, following the same steps as for the aerosol filters. Additionally, up to
100 mL of bulk seawater samples and SML samples were fixed, stained, filtered and the resulting filters mounted on slides and
stored at -20 °C. Details regarding the exact seawater volume passed through each filter are available in Tables S1 and S2.

For enumeration, a fluorescence microscope (BX60, Olympus Corporation, Japan) equipped with a UV filter set (excitation
wavelength approximately 365 nm) was used. Counting of positively stained cells was performed under 1000x magnification
and using the FIJI software (Schindelin et al., 2012). Positively stained cells were counted in 20 random fields per sample,
corresponding to an area of 0.3 mm$^2$ of the filter, to ensure reliable counting statistics.

## 2.3 Inorganic ion analysis

The extracted aerosol and seawater samples were analyzed for their ionic composition using a Dionex Aquion IC system
(Thermo Fischer Scientific, US). An AS22 column (eluent 20 mM methanesulfonic acid pumped at 1 mL min$^{-1}$) and a CS12A
column (eluent 1.5 mM NaHCO$_3$ and 4.5 mM Na$_2$CO$_3$ pumped at 1.2 mL min$^{-1}$) were used to determine the concentrations of
major anions (chloride, Cl$^-$; sulphate, SO$_2^{4-}$) and cations (sodium, Na$^+$; potassium, K$^+$; magnesium, Mg$^{2+}$; calcium, Ca$^{2+}$),
respectively. The injected volume was 25 μL. Seawater samples were diluted 1:20 with ultrapure water to match the analytical
range of the aerosol extracts.

To ensure analytical quality, certified reference samples (QC DWB, Eurofins Miljø Luft A/S, Denmark) were used for checks.
Systematic errors were less than 2% for all ionic components, except for Ca$^{2+}$, which had less than 3% error. Random errors
were approximately 0.1% for samples analyzed after the R/V *Oceania* campaign and slightly higher after the R/V *Electra*
campaign (0.2%, 1.3%, 0.5%, 0.2%, 3.4% and 3.2% for Na$^+$, K$^+$, Mg$^{2+}$, Ca$^{2+}$, Cl$^-$, SO$_4^{2-}$, respectively).

The limits of detection (LOD), defined as three standard deviations of the ultrapure water blanks, were 0.006, 0, 0.001, 0.057
and 0.052 mg L$^{-1}$ for Na$^+$, K$^+$, Mg$^{2+}$, Ca$^{2+}$, and Cl$^-$, respectively, for samples collected during the R/V *Oceania* campaign.
For samples collected during the R/V *Electra* campaign, the LODs were 0.004, 0.003, 0.002, 0.008, 0.012 and 0.001 mg L$^{-1}$
for Na$^+$, K$^+$, Mg$^{2+}$, Ca$^{2+}$, Cl$^-$ and SO$_2^{4-}$, respectively. The limits of quantification (LOQ), defined as ten standard deviations
of the ultrapure water blanks, were 0.021, 0, 0.002, 0.192, and 0.174 mg L$^{-1}$ for Na$^+$, K$^+$, Mg$^{2+}$, Ca$^{2+}$, and Cl$^-$, respectively,



for samples collected during the R/V *Oceania* campaign. For samples collected during the R/V *Electra* campaign, the LOQs were 0.015, 0.008, 0.007, 0.027, 0.040, and 0.005 mg L$^{-1}$ for Na$^+$, K$^+$, Mg$^{2+}$, Ca$^{2+}$, Cl$^-$, and SO$_2^{4-}$, respectively.

## 2.4 Calculation of enrichment factors

We normalized the cell concentration per mL of extract from the filter samples and corresponding seawater samples (referred to as $X$) by their respective Na$^+$ concentrations to determine the enrichment factors (EFs) of aerosolized cells relative to seawater.

The formula for calculating the EF is as follows:

$$\mathrm{EF} = \frac{\left(\frac{\mathrm{X}}{\mathrm{Na}}\right)_{\mathrm{aerosol}}}{\left(\frac{\mathrm{X}}{\mathrm{Na}}\right)_{\mathrm{seawater}}} \tag{1}$$

In this equation, $\left(\frac{X}{Na}\right)_{\mathrm{aerosol}}$ is the ratio of cell concentration ($X$) to Na$^+$ concentration in the aerosol samples and $\left(\frac{X}{Na}\right)_{\mathrm{seawater}}$ is the same ratio in the corresponding seawater samples.

## 2.5 DNA analysis

The DNA extraction and sequencing procedure used in this study are detailed in Zinke et al. (2024b). Briefly, nucleic acids were extracted from seawater and aerosol filter samples using the Plant Easy Mini kit (Qiagen, Germany) with a final elution volume of 25 µL and amplified the V3-V4 region of the 16S rRNA gene through a triple polymerase chain reaction (PCR) using the Bac341F primer 5'-CCT ACG GGN GGC WGC AG-3' and the Bac805R primer 5'-GAC TAC HVG GGT ATC TAA TCC-3' (Herlemann et al., 2011; Hugerth et al., 2014). These primers were optimized for bacteria, and might not amplify

archaea as effectively (Hugerth et al., 2014). The 16S rRNA gene amplification was performed at the Department of Biology, Aarhus University, Denmark, and followed a modified Illumina protocol (16S Metagenomic Sequencing Library Preparation, Part 15044223 Rev. B). A detailed desciption of the triple PCR amplification can be found in the supplementary section S2. It should be noted that the triple PCR may have introduced biases due to over-amplification of certain taxa, due to differences in template abundances, differences in amplification efficiency (Polz and Cavanaugh, 1998) or inhibition of amplification

by self-annealing of the most abundant templates (Suzuki and Giovannoni, 1996). The nf-core/ampliseq workflow version 2.3.2 (Straub et al., 2020) was used to analyse the sequencing data (see section S3 for a more detailed description of the workflow). The R Phyloseq package (McMurdie and Holmes, 2013) and the R Vegan package (Oksanen, 2010) were used to analyze and visualize the processed data. 175 amplicon sequence variant (ASVs) were identified as contaminants and removed from further analysis using the decontam package (Davis et al., 2017). A list of those ASVs is provided in Table S3. The R

breakaway package (Willis and Bunge, 2015) was used to estimate the alpha diversity and similarities between between the seawater and chamber aerosol samples were assessed using non-metric multidimensional scaling analysis (NMDS) based on the Bray–Curtis similarity index. An ANOSIM test was conducted (Clarke, 1993) to test for significant differences between the different sample types. Taxonomic trees were generated using PhyloT and iTOL (Letunic and Bork, 2021). To investigate the bacteria that were more likely to be aerosolized or remain in the seawater, aerosolization factors (AF) were calculated as



the mean relative abundance in the aerosols in the head space of the sea spray simulation chamber ($C_{\mathrm{SSC}}$) divided by the mean relative abundance in the seawater ($C_{\mathrm{seawater}}$):

$$\mathrm{AF} = \frac{C_{\mathrm{SSC}}}{C_{\mathrm{seawater}}}. \tag{2}$$

### 2.6 Aerosol particle measurements

During both campaigns, the size distribution of the aerosols produced in the chamber was measured using a custom-built dif-
ferential mobility particle sizer (DMPS), consisting of a Vienna-type differential mobility analyzer (DMA) and a condensation particle counter (CPC, model 3772, TSI, USA), and a white-light optical particle size spectrometer (WELAS 2300 HP sensor and Promo 2000 H, Palas GmbH, Germany, hereafter called OPSS). The DMPS measured particles with electrical mobility diameters between 0.015 and 0.906 µm distributed at a flow rate of $1\,\mathrm{L\,min^{-1}}$, while the WELAS measured particles with optical diameters between 0.150 and 10 µm at a flow rate of $5\,\mathrm{L\,min^{-1}}$. The size distributions were combined at 350 nm. A
more detailed description of the aerosol sizing instrumentation is described in Zinke et al. (2024c). During the *Electra* campaign, we conducted additional concentration and fluorescence measurements of coarse particles (CP), fluorescent particles (FP), and fPBAP with optical diameters larger than 0.8 µm using a single-particle Multiparameter Bioaerosol Spectrometer (MBS, University of Hertfordshire, U.K.). The MBS had a sample flow of $0.3\,\mathrm{L\,min^{-1}}$ and a sheath flow of $1.55\,\mathrm{L\,min^{-1}}$. The MBS detects and sizes particles using a low-power laser with a wavelength of 635 nm to detect and size individual particles.
A xenon flash-lamp with a wavelength of 280 nm excites fluorescence which is recorded in equidistant 8 channels within the spectral range of 305 and 655 nm. Further details about the MBS can be found in Ruske et al. (2017). The particular MBS data processing used here is described in Freitas et al. (2022), which is based on a similar set-up of the MBS connected to the sea spray chamber. In brief, if a particle's fluorescence signal exceeds three times the standard deviation ($3\gamma$) of the background signal it is classified as a fluorescent particle (FP) and if it exceeds $9\gamma$ in any of the fluorescence channels it is classified as
highly fluorescent particles (HFP). This threshold has been used in previous studies by Savage et al. (2017) and Freitas et al. (2022). If the maximum fluorescent signal is observed at 364 nm, particles are classified as marine fPBAP (Freitas et al., 2022). Freitas et al. (2022) showed that particles with a maximum at 364 nm were only present in real seawater but were absent after the filtration of the seawater.

To ensure dry air for aerosol sampling, the aerosol particle-laden air from the sea spray simulation chamber was passed
through a Nafion dryer (MD-700-36F, Perma Pure, USA) located in front of the MBS. The dryer was supplied with a sheath flow of dry, particle-free air at a flow rate of $10\,\mathrm{L\,min^{-1}}$. The relative humidity and temperature of the sampled air were monitored with sensors (HYTELOG-USB, B+B Thermo-Technik GmbH) mounted in front of the MBS sampling inlets. The average RH was $19.1 \pm 3.0\%$ and the average temperature of the dried chamber air was $26.4 \pm 1.6°\mathrm{C}$.

A schematic of the experimental set-up is depicted in Fig. 1.




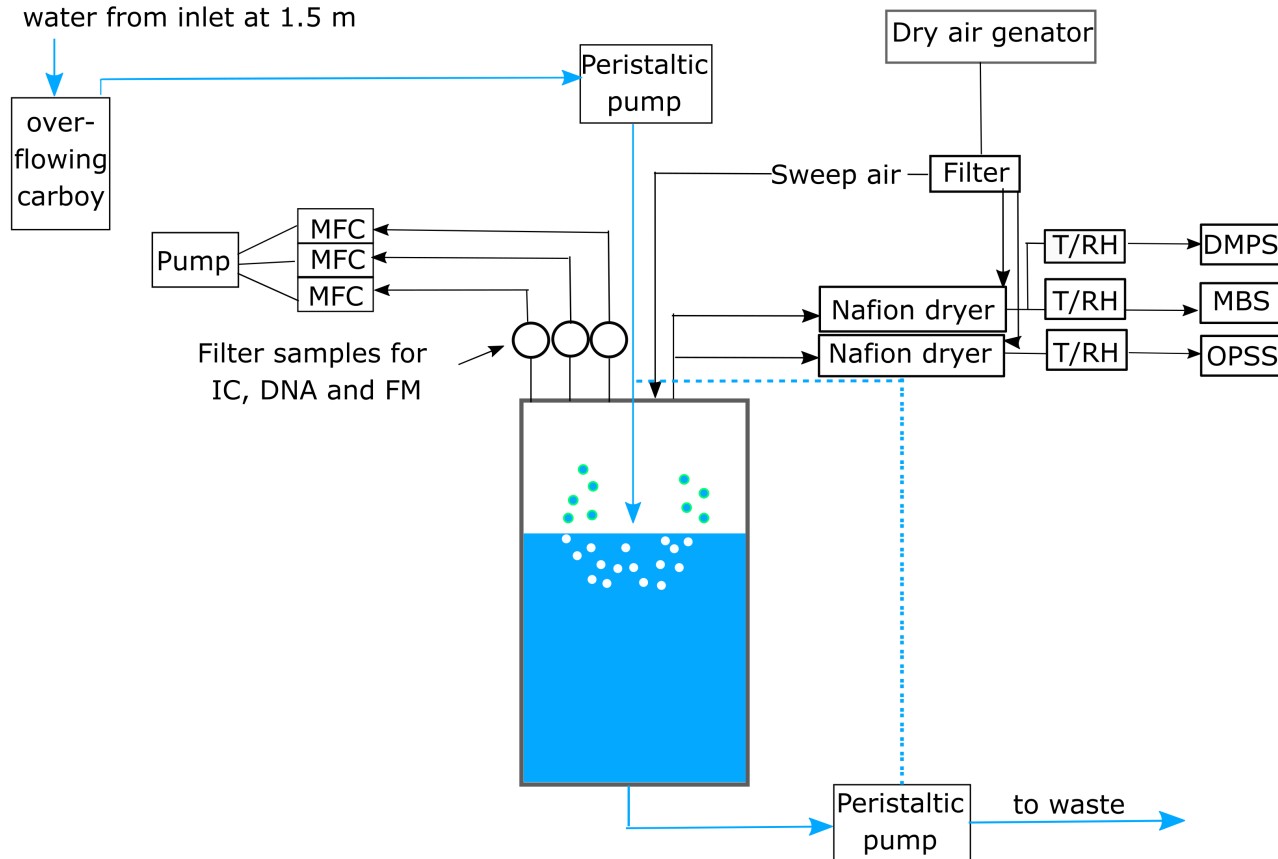

**Figure 1.** Schematic of the experimental set-up. The seawater in the chamber was constantly plunged and refilled from the ship's seawater intake. The sea spray aerosols (SSA) produced in the chamber were sampled onto three filters for bacteria enumeration using fluorescence microscopy (FM), ion chromatography (IC) and DNA analysis. The flow through each filter was controlled using mass flow controllers (MFC). Aerosol number and size were measured with a differential mobility particle sizer (DMPS) and an optical particle size spectrometer (OPSS). Additional online measurements of fluorescent particles were conducted using a mulitparameter bioaerosol sensor (MBS). Prior to sampling the aerosol-laden air was dried using a nafion dryer and the temperature ($T$) and relative humidity (RH) in the sampling line was monitored using a T/RH sensor. The dashed blue line indicates the water flow when the chamber was operated in closed mode.

### 2.7 Aerosol eddy covariance flux measurements

Additional data were obtained from flux measurements conducted on Östergarnsholm island, as described in detail in Zinke et al. (2024c). In brief, aerosol eddy covariance fluxes were measured at 12 m above sea level using an ultrasonic anemometer (Gill HS, Gill Instruments Ltd, UK) that recorded the three-dimensional wind speed and atmospheric temperature at a frequency of 20 Hz. Fluctuations in $H_2O$ and $CO_2$ were recorded at the same frequency using a Licor 7500A (Li-Cor Environmental Ltd, UK). The concentration of ambient aerosols with diameters $0.25 < D_p < 2.5\,\mu m$ was measured at ambient relative humidity



with a time resolution of 1 s using an optical particle counter (OPC, Model 1.109, Grimm Aerosol Technik GmbH, Germany). For detailed information about the flux calculations, we refer the reader to Zinke et al. (2024c).

# 3 Results and discussion

## 3.1 Sea spray chamber experiments onboard R/V Electra

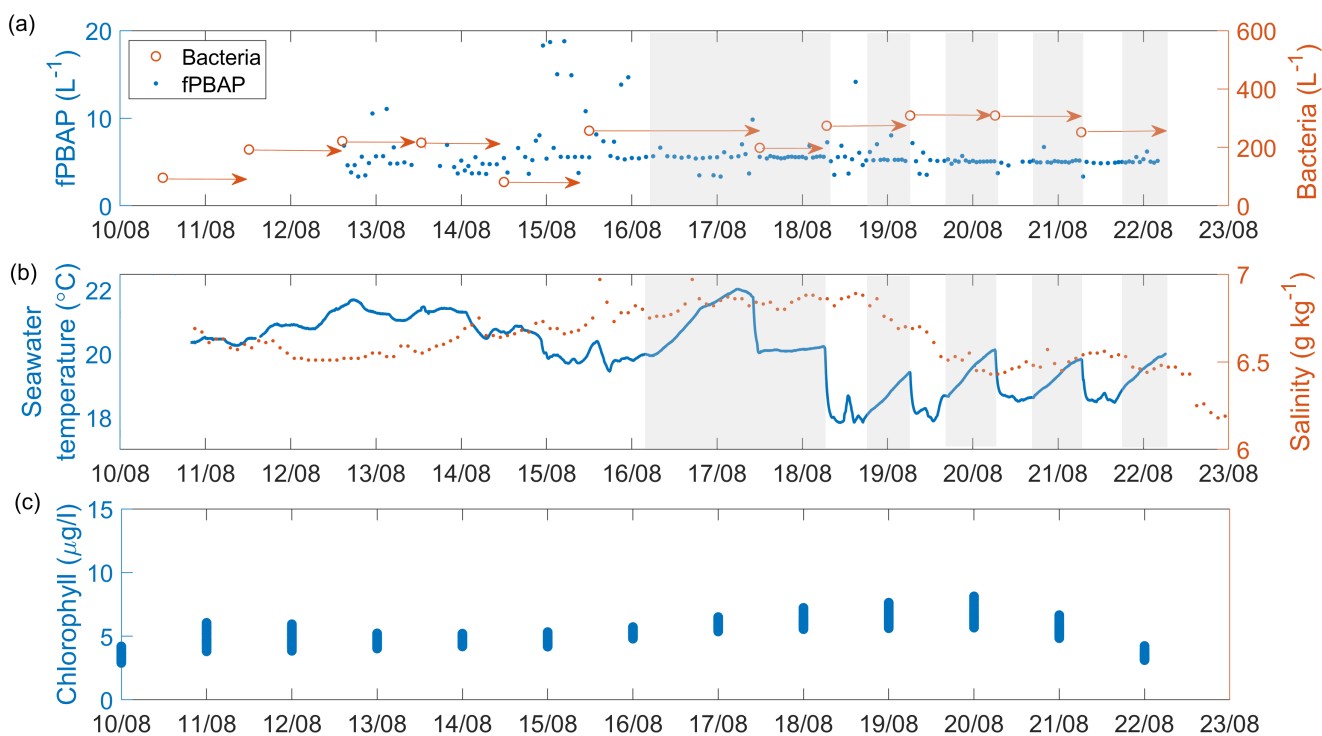

**Figure 2.** Time series of (a) fluorescent primary biological aerosol particles (fPBAP) and airborne bacteria concentration measured in the head space of the chamber, (b) seawater temperature and salinity and (c) concentration of chlorophyll-*a* (from re-analysis data) in the seawater measured during the *Electra* campaign. The arrows in panel (a) mark the duration of the filter collection. Times when the chamber was operated in closed mode are shaded in grey.

A time series of seawater properties such as temperature, salinity, chlorophyll-*a* along with the concentrations of MBS-derived fPBAP and airborne bacteria determined from filters measured from the sea spray chamber experiments during the R/V *Electra* campaign are shown in Fig. 2. The average seawater temperature inside the chamber remained stable initially around $20\pm1.1°C$ and later around $18\pm1.1°C$ towards the end of the campaign. However, during closed mode operations, the temperature rose by an average $1.5\pm0.4°C$ due to the lack of fresh seawater input. Throughout the campaign, seawater salinity

remained constant at $6.7\pm0.1\,\mathrm{g\,kg^{-1}}$.



The average chlorophyll-*a* concentration, derived from re-analysis data, remained stable at $5.2 \pm 1.1$,mg,m$^{-3}$ throughout the *Electra* campaign. Chlorophyll-*a* serves as an indirect indicator of biomass in surface seawater and has been utilized here to gauge the emission of organic matter and bacteria from ocean surfaces. However, recent studies by Quinn et al. (2014) and Freitas et al. (2022) suggest that correlating chlorophyll-*a* concentrations directly with organic matter and bacteria emissions

may not be straightforward. Despite the consistent daily averages during the *Electra* campaign, significant correlations between chlorophyll-*a* concentrations and PBAP emissions were not observed. The growth of plankton biomass in the Baltic Sea follows a distinct seasonal cycle, characterized by three prominent peaks (Wasmund et al., 1996; Stoń-Egiert and Ostrowska, 2022; Skjevik et al., 2022). These peaks typically occur during spring (March/April), when diatoms and dinoflagellates bloom; summer (July/August), marked by the proliferation of large filamentous cyanobacteria; and autumn (September/October), when

diatoms bloom again (the timing of the blooms noted here are general averages and there are on occasion anamolies). The R/V *Oceania* campaign, conducted in May, fell between the spring and summer blooms, whereas the R/V *Electra* campaign, carried out in August, coincided with the late summer bloom. Figure S2 in the supplement displays a yearly time series of the re-analyzed chlorophyll-*a* concentration in the vicinity of Östergarnsholm.

The average concentration of bacterial cells, as determined by fluorescence microscopy was $4.5 \cdot 10^5 \pm 3.7 \cdot 10^5\,\mathrm{mL}^{-1}$ in the

bulk seawater (SW) and $3.2 \cdot 10^5 \pm 2.6 \cdot 10^5\,\mathrm{mL}^{-1}$ in the SML (see Fig. S3). Surprisingly, the number of bacteria in the SML samples did not significantly differ from those in the bulk seawater (Pearson's correlation coefficient r=0.94, p=0.002, n=7). This contrasts with previous studies that have reported higher bacterial concentrations in the SML (Sieburth et al., 1976; Aller et al., 2005; Rastelli et al., 2017).

The average concentration of bacteria in the chamber's headspace increased from $178.4 \pm 45.9\,\mathrm{cells\,L}^{-1}$ to $241.0 \pm 77.5\,\mathrm{cells\,L}^{-1}$

when the plunging jet flow rate was raised from $1.3\,\mathrm{L\,min}^{-1}$ to $2.6\,\mathrm{L\,min}^{-1}$. In contrast, the concentration in blank filter samples was four orders of magnitude lower, with only $4.5 \cdot 10^2 \pm 2.7 \cdot 10^2$ cells per blank filter compared to $1.4 \cdot 10^6 \pm 8.8 \cdot 10^5$ cells per aerosol filter. Furthermore, increasing the plunging jet flow rate resulted in higher concentrations of CP and FP ($4865 \pm 1291\,\mathrm{CP\,L}^{-1}$ at $1.3\,\mathrm{L\,min}^{-1}$ compared to $7316 \pm 1698\,\mathrm{CP\,L}^{-1}$ at $2.6\,\mathrm{L\,min}^{-1}$ and $422 \pm 122\,\mathrm{FP\,L}^{-1}$ at $1.3\,\mathrm{L\,min}^{-1}$ compared to $646 \pm 153\,\mathrm{FP\,L}^{-1}$ at $2.6\,\mathrm{L\,min}^{-1}$, respectively). Surprisingly, the concentration of fPBAP did not show a signif-

icant increase ($5.4 \pm 2.4\,\mathrm{L}^{-1}$ at $1.3\,\mathrm{L\,min}^{-1}$ compared to $5.7 \pm 1.9\,\mathrm{L}^{-1}$ at $2.6\,\mathrm{L\,min}^{-1}$). Notably, fPBAP represented only a minor fraction of all FP ($1.4 \pm 2.7\%$) and an even smaller proportion of CP ($0.15 \pm 0.38\%$).

These ratios are higher than those reported in a previous study by Freitas et al. (2022), who conducted similar experiments in the Baltic Sea near Gotland in June 2018 using the same experimental setup and reported fPBAP/CP ratios of 0.05%. One potential factor influencing the airborne fPBAP concentration could be seasonal differences in the abundance of biological and

fluorescent matter in surface water between August 2021 and June 2018.

The fPBAP estimates obtained in this study were nearly two orders of magnitude lower than the cell abundance estimates obtained from fluorescence microscopy. This discrepancy may be attributed to two main factors. Firstly, the limitations of the MBS instrument, which only detects particles with diameters larger than 0.8,μm, could lead to undersampling of smaller bacterial cells, viruses, and cell fragments. Secondly, while the MBS provides an estimate of particle-attached bacteria and

bacteria emitted as agglomerates, as they are present in the ambient atmosphere, fluorescence microscopy of the extracted





filters only provides estimates of single bacterial cell numbers. This distinction is significant because cell clusters are likely to break up during the sonication step (Zinke et al., 2024b).

### 3.1.1 Enrichment factors of bacteria in chamber air and SML compared to the underlying bulk seawater

To assess the EF of bacteria in the sea spray simulation chamber aerosol relative to the underlying seawater, we normalized the estimated bacteria abundances by the respective sodium concentrations. This normalization adjusts for changes in the SSA emission flux within the chamber, which is influenced by experimental conditions such as the plunging jet speed. A time series of the bacteria and sodium concentrations in both seawater and chamber aerosol samples is provided in Fig. S3. The mass fractions of sodium in the aerosol and seawater samples remained relatively constant throughout both campaigns, with average values of approximately $30.5 \pm 2.5\%$ and $34 \pm 1\%$ in the aerosol, and $32.6 \pm 0.3\%$ and $32 \pm 0.1\%$ in the seawater, during the R/V *Oceania* and R/V *Electra* campaigns, respectively (see Fig. S4 in the supplement).

The derived EFs ranged from 13 to 488 (mean $125.4 \pm 143.3$) and decreased from $262.6 \pm 162.4$ to $47.0 \pm 36.2$ as the plunging jet flow rate was increased from $1.3 \, \mathrm{L \, min^{-1}}$ to $2.6 \, \mathrm{L \, min^{-1}}$. This decrease could be attributed to the higher jet flow rate potentially preventing the formation of a SML, which typically contains higher concentrations of bacteria and organic material.

Contrary to previous studies (Sieburth et al., 1976; Aller et al., 2005; Rastelli et al., 2017), we observed no significant enrichment of bacteria in the SML compared to the underlying SW. The EF between the SML and SW ranged from 0.34 to 1.74, with a mean of $0.92 \pm 0.51$. One possible explanation for this lack of enrichment is the average wind speed of $6.7 \pm 2.5 \, \mathrm{m \, s^{-1}}$ encountered during the *Electra* campaign, which may have been high enough to prevent bacterial enrichment in the SML due to wind-induced mixing. We observed a negative correlation (r=-0.87, p=0.01) between wind speed and SML EF, with the highest SML EF occuring on the day with the lowest wind speed (see also Fig. S5). This result is consistent with the findings of Rahlff et al. (2017), who found that wind speeds above $4.1 \, \mathrm{m \, s^{-1}}$ prevent bacterial enrichment in the SML. Notably, several studies have reported that the SML can persist at wind speeds above $6.6 \, \mathrm{m \, s^{-1}}$, and enrichment of organic matter has been observed at wind speeds up to $10 \, \mathrm{m \, s \, m \, s^{-1}}$ (e.g., Carlson, 1983; Wurl et al., 2011). It is important to recognize that the SML is operationally defined and likely persists even at moderate to high wind speeds. Despite this persistence, there are clearly periods, especially at higher wind speeds, when bacteria are not enriched in the SML. Another possible explanation for the lack of enrichment in our study could be the dilution of the sample with subsurface water during glass-plate sampling.

### 3.1.2 Comparison of bacterial abundance estimates and enrichment factors with previous mesocosm studies

In early investigations, Blanchard and Syzdek (1972) conducted mesocosm experiments using a suspension of *Serratia marcescens* in distilled water in order to observe and quantify dynamics in EF production. They observed significantly higher EFs in the initial jet drops, with values up to 1200, compared to the final jet drops with EFs of 8. Additionally, Blanchard and Syzdek (1982) reported EFs ranging between 10 and 20 for film drops, while Cipriano (1979) conducted a separate mesocosm study using a suspension of *S. marinorubra* in seawater from Long Island and reported EFs ranging from 50 to 100 for film drops. While we did not distinguish between film and jet drops in the current study, we can expect that the lower salinity of the Baltic





seawater lead to a presence of larger bubbles that tend to break up into film drops (Zinke et al., 2022), which might explain

why the EFs observed in the current study fall closer to the estimate from Cipriano (1979).

Other mesocosm studies have used culture based approaches to enumerate the abundance of airborne bacteria from the Baltic Sea and Arctic Ocean, observing lower concentrations than our study (Marks et al., 2001; Hultin et al., 2011; Fahlgren et al., 2015). However, it should be considered that not all bacteria form visible colonies (less than 1% of bacteria are culturable), which could explain the lower concentrations reported in these studies. Marks et al. (2001), who conducted mesocosm ex-

periments with Baltic seawater from the Bay of Gdansk in July 1997 and March 1998, estimated bacterial EFs of mesophilic and psychrophilic bacteria ranging between 37-2545 and 14-585, respectively. Hultin et al. (2011), who conducted mesocosm experiments in the Baltic Sea in May-June and September 2005, provided an upper estimate of the total airborne bacteria concentration of $10^4 - 10^6$ cells m$^{-3}$ (or $10^1 - 10^3$ cells L$^{-1}$) by assuming that the transport efficiency of all bacteria is equal to that of the colony forming units (CFU). This estimate falls approximately within the same range as the concentrations obtained

from the current study.

Other mesocosm studies that reported airborne bacteria concentrations comparable to the estimates obtained from the current study were conducted by Aller et al. (2005), Rastelli et al. (2017) and Zinke et al. (2024b). Aller et al. (2005) used inverted funnels and frits to generate SSA directly from seawater on a floating catamaran in June-September 2003 in the Long Island Sound, New York. They enumerated DAPI-stained bacteria with fluorescence microscopy and reported EFs of approximately

10. However, frits have been shown to produce a narrower bubble size spectrum than the spectrum observed in breaking waves with a bias towards smaller bubbles (Stokes et al., 2013). These smaller bubbles tend to produce jet drops, possibly explaining the lower EF compared to our study.

Rastelli et al. (2017) conducted mesocosm experiments in June-July 2006 in the North-Eastern Atlantic (close to Ireland) using a plunging jet sea spray simulation chamber. Samples were collected with a five stage Berner impactor and analyzed

with fluorescence microscopy to estimate the bacteria abundance in the aerosol. They reported size-dependent enrichment of bacteria in SSA with EF of approximately 45 for particles with diameters less than 1.2 µm and significantly lower enrichment for particles larger than 1.2 µm. Similarly, Michaud et al. (2018) determined the abundance of airborne bacteria from wave channel experiments at the Scripps Pier in San Diego, USA using flow cytometry and reported EFs of approximately 11 in SSA compared to the bulk seawater.

Recently, in a mesocosm study conducted in June-July 2022 the North-Eastern Atlantic, Zinke et al. (2024b) derived an average EF of $48.6 \pm 35.6$ (ranging between 9 and 158). While this study used a similar approach to that employed in the current study, it used a smaller sea spray simulation chamber that was operated in closed mode, meaning the water was not replaced over the duration of each 48 h experiment. As a result of this study, selective growth in the seawater was observed that might have impacted the enrichment of certain bacteria taxa in the chamber air. In the current study, we have employed an

approach where the seawater is continuously replaced to prevent such selective growth, except for periods when the ship had to leave its anchored position close to Östergarnsholm.

It is important to note that, among the studies discussed above, Rastelli et al. (2017) and Zinke et al. (2024b) are the only ones that normalized their EF using the sodium concentration. This is important because it is essential to consider that the





emission fluxes of bacteria or PBAP obtained from mesocosm experiments can be influenced by various factors, such as the

biogeochemical properties of the seawater (e.g., salinity, seawater temperature, presence of surfactants) and the experimental

setup (e.g., bubble-production method, bubbling rate, proximity of sampling ports to the water surface, chamber size, and

closed or flow-through operation mode). Therefore, caution should be exercised when comparing results from experiments

with different setups. By normalizing the EF using the sodium concentration, it becomes possible to account for changes in

SSA emission fluxes. To facilitate meaningful comparisons across different studies, we strongly advocate for normalizing the

EF using relevant ionic concentrations, such as sodium. This approach will help ensure a more robust and accurate assessment

of the enrichment and emission of bacteria and fPBAP from seawater to the atmosphere. A summary table of the above

discussed studies can be found in Zinke et al. (2024b).

### 3.1.3 Estimation of marine bacteria and fPBAP fluxes from the chamber experiments

Emission fluxes of bacteria and fPBAP (per $\mathrm{m}^{-2}\,\mathrm{s}^{-1}$) were derived using two independent approaches: In the first approach,

bacteria emission fluxes were derived by multiplying the EFs derived in this study with mass emissions estimates from existing

SSA parameterizations from Mårtensson et al. (2003), Salter et al. (2015) and Zinke et al. (2024c):

$$F_{N,bacteria} = EF \frac{[\mathrm{Na^+}]_{\text{seawater}}}{[\text{sea salt}]_{\text{seawater}}} \frac{[\text{bacteria}]_{\text{seawater}}}{[\mathrm{Na^+}]_{\text{seawater}}} \rho \frac{\pi}{6} \int\limits_{D_{\min}}^{D_{\max}} \lceil D^3 \frac{dF_{N,X}}{d\log D} \rceil \, d\log D \tag{3}$$

where $\frac{[Na^+]_{\text{seawater}}}{[sea\ salt]_{\text{seawater}}}$ is the fraction of sodium mass to the total sea salt mass in seawater (here we used the mass fraction of

sodium in the aerosol, 0.32, that was measured during the *Electra* campaign), $\frac{[\text{bacteria}]_{\text{seawater}}}{[Na^+]_{\text{seawater}}}$ is the ratio of bacteria abundance

to sodium mass measured per mL of seawater, $\rho$ is the sea salt aerosol density (here we used used $2.017\,\mathrm{g\,cm^{-3}}$, Zieger

et al., 2017), $\frac{dF_{N,X}}{dlogD}$ is the size resolved number emission from one of the sea salt source parameterizations (so X is either

Mårtensson, Salter or Zinke). The mass fluxes were integrated over a size range of $0.02 < D_p < 2.8\,\mu\mathrm{m}$, which corresponds to

the range in which the Mårtensson et al. (2003) parameterization is valid. The parameterizations of Mårtensson et al. (2003)

and Salter et al. (2015) were derived for salinities 33 and 35 $\mathrm{g\,kg^{-1}}$, respectively. Since the current study was conducted under

brackish conditions ($S \sim 6.7\,\mathrm{g\,kg^{-1}}$), a correction factor $\frac{6.7}{(33\ \text{or}\ 35)}$ had to be applied for these two parameterizations. This

correction factor is based on the assumption of a linear relationship between salinity and SSA volume, which is supported by

the findings of Zinke et al. (2022). On the other hand, the parameterization by Zinke et al. (2024c) was derived for the exact

conditions of the current study. While the parameterizations by Mårtensson et al. (2003) and Salter et al. (2015) are based on

purely inorganic sea salt, the parameterization Zinke et al. (2024c) was derived for SSA containing organics. Unfortunately,

the amount of organics in the SSA is unknown.

The second approach is based on a comparison of the particle concentration measured from the chamber experiments with

in-situ eddy covariance fluxes measured on Östergarnsholm island (Zinke et al., 2024c). From this comparison we derived

a chamber-specific scaling factor which allows us to scale the airborne bacteria counts to bacteria fluxes in units of cells per

$\mathrm{m}^{-2}\mathrm{s}^{-1}$. Using this scaling factor we estimated an average bacteria flux of $45.9\pm13.4\,\mathrm{cells\,m^{-2}s^{-1}}$ (range 16-63 $\mathrm{cells\,m^{-2}s^{-1}}$)





**Figure 3.** Estimation of single bacteria cell and fPBAP emission fluxes at different wind speeds using two different approaches: The red and blue line show fPBAP and bacteria fluxes that were estimated by multiplying the ratio of bacteria or fPBAP to all aerosols with the wind speed dependent parameterization from Zinke et al. (2024c). The green, yellow and purple lines were estimated by multiplying the enrichment factors with the mass flux estimated from the parameterizations of Zinke et al. (2024c), Salter et al. (2015) and Mårtensson et al. (2003), respectively. The dashed yellow and purple lines show the estimates based on Salter et al. (2015) and Mårtensson et al. (2003) but accounted for the difference in salinities between those parameterizations and the current study. The lines are the mean flux estimates while the shaded areas/ error bars indicate the range of the minimum to maximum flux estimates. All number/ mass fluxes were integrated over a size range of $0.02 < D_p < 2.8\,\mu m$, which corresponds to the range in which the Mårtensson et al. (2003) parameterization is valid.





and an average fPBAP flux of $1.12 \pm 0.47\,\mathrm{fPBAP\,m^{-2}s^{-1}}$ (range 0.6-6.1 fPBAP m$^{-2}$s$^{-1}$) at the given experimental conditions. Furthermore, by multiplying the wind-speed dependent parameterization derived in Zinke et al. (2024c) with a factor of 0.00086 (mean, range 0.0003 - 0.0013), which corresponds to the fraction of bacteria cells determined from fluorescence microscopy to the total aerosol concentration (integrated over the size range between 0.02 to 2.8 μm), we were able to derive bacteria emission fluxes for different wind speeds (see Fig. 3). A two-sample Kolmogorov-Smirnov test (Massey Jr, 1951) revealed no

significant differences between the estimates from the two different approaches used in this study (p=0.99) at a significance level of 5%. Similar to the emission flux of bacteria we estimated the wind dependent emission fluxes of fPBAP by multiplying with a factor of $2.209 \cdot 10^{-5}$ (mean, range $9.945 \cdot 10^{-6}$–$4.627 \cdot 10^{-5}$). The estimated fPBAP emissions were more than an order of magnitude lower than the bacteria emission fluxes, likely due to different measurement approaches (single cells from fluorescence microscopy versus particle-attached cells or cell agglomerates from the MBS measurements) as discussed in

section 3.1.

Only a few other studies have attempted to estimate the emission flux of bacteria or PBAP. Mayol et al. (2014), who conducted ambient measurements in the North-Atlantic, derived a flux of prokaryotes (defined as cells with diameters <1 μm) of 42 cells m$^{-2}$s$^{-1}$ by multiplying the abundance in seawater with the SSA source function from Andreas (1998). They further estimated a deposition flux of 49 prokaryotic cells m$^{-2}$ s$^{-1}$ by multiplying the atmospheric concentration of prokaryotes with

the settling velocity, which depends on particle size and density. By subtracting the deposition flux from the emission flux, they derived a net flux of -6.49 prokaryotes m$^{-2}$s$^{-1}$. Similarly, Mayol et al. (2017) estimated the total emission flux of prokaryotes over the subtropical and tropical oceans (Atlantic, Pacific and Indian Ocean) to be $1 \cdot 10^3 - 2 \cdot 10^6$ cells m$^{-2}$ day$^{-1}$ (or 0.01-23 cells m$^{-2}$s$^{-1}$) and a deposition flux of up to $6 \cdot 10^6$ cells m$^{-2}$ day$^{-1}$ (or 69.4 cells m$^{-2}$s$^{-1}$). Hu et al. (2017) used the same approach to estimate bacteria fluxes from ambient measurements in the Kuroshio extension and reported values on the order of

$1 \cdot 10^2$ cells m$^{-2}$s$^{-1}$. Hu et al. (2017) further estimated the wind speed dependent emission fluxes of bacteria using enrichment factors and seawater concentrations of bacteria (see their Fig. 3a and supplemental table S1). It should be noted that the studies by Mayol et al. (2014), Mayol et al. (2017) and Hu et al. (2017) were conducted under different conditions than the current studies (i.e. in high salinity oceans with different wave states and varying distances to land). Even so, the emission flux estimate obtained from the current study falls within the same range as the estimates from those studies. These previous studies suggest

that the deposition flux of marine prokaryotes outweighs their emission flux (Mayol et al., 2017; Hu et al., 2017), implying that the ocean acts as a sink rather than a source of bacteria (except at high wind speeds > 8 m s$^{-1}$, Hu et al., 2017). However, these studies only provide *a posteriori* estimates of the fluxes, which might also be affected by terrestrial sources due to the proximity to land. Combining eddy covariance systems with high-frequency online fPBAP measurements would allow direct measurements of fPBAP fluxes and could help determine whether the oceans are actually a sink or a small source of fPBAP.

### 3.1.4 Bacteria community composition in the seawater and chamber aerosols

Using 16S rRNA sequencing we have investigated the bacterial community composition in three different sample types: bulk seawater (SW), SML and nascent SSA generated in a sea spray simulation chamber (SSC) during both campaigns (see Fig. 4). A total of 3033 ASVs were shared between the seawater and the aerosol samples from both campaigns. There were 773 ASVs





unique to the seawater from the R/V *Electra* campaign, 366 ASVs unique to the seawater from the R/V *Oceania* campaign,

ASVs unique to the SSC samples from the R/V *Oceania* campaign, and 102 ASVs unique to the SSC samples from the R/V *Electra* campaign (see Fig. S6). A comparison of the communities in the SW, SML and SSC using the ANOSIM statistical test revealed distinct differences between the communities in SSC and SW (with a dissimilarity value of r=0.76, p=0.002 for the R/V *Oceania* campaign; r=1, p=0.002 for the R/V *Electra* campaign) and SSC and SML (r=0.67, p=0.05 for the R/V *Oceania* campaign; r=1, p=0.004 for the *Electra* campaign). This is surprising given that the bacteria sampled from the

head space of the chamber were expected to originate from the seawater. One possible explanation for this could be selective aerosolization of certain taxa. Fahlgren et al. (2015) also reported that bacterial communities in aerosols differed greatly from corresponding seawater communities. Their conclusion was that some taxa were selectively enriched in aerosols, sometimes even entirely removed from the seawater samples, while others were barely aerosolized at all. Another possible explanation could be the presence of free DNA in the head space of the chamber that can pass through the HEPA-filters used to flush

the head space with particle free air (Zinke et al., 2024b). No significant difference was observed between SML and SW for both campaigns (r=-0.3, p=0.88 for the R/V *Oceania* campaign; r=0.14, p=0.18 for the *Electra* campaign), suggesting that the bacteria community in the SML was well mixed with the underlying bulk water.

  Regarding alpha diversity, the samples from the R/V *Oceania* campaign showed slightly less diversity than those from the R/V *Electra* campaign (see Fig. S7a). In line with this observation, previous studies at different study sites have reported

seasonal cycles in the bacterial diversity with a minimum during spring and a maximum during late summer and autumn that are governed by environmental parameters such as seawater temperature, day length and nutrient availability (e.g. Fuhrman et al., 2006; Andersson et al., 2010). In terms of beta diversity, the samples from the two campaigns formed distinct clusters (see Fig. 7b). The SSC samples clustered separately from the SW samples for both campaigns. In the R/V *Electra* campaign, SML samples showed slightly less diversity than SW samples in terms of alpha diversity but were closely clustered in terms of

beta diversity. In the R/V *Oceania* data, no distinct difference between SW and SML samples was observed in terms of alpha or beta diversity.

  We did not observe a significant enrichment of certain taxa in the SML when compared to the SW (see Fig. S8) which is in line with the findings that there was no enrichment in terms of the total cell abundance either. It is possible that high wind conditions prevented the formation of a distinct bacterioneuston community in the SML due to constant mixing of the SML

with the underlying bulk seawater as discussed in section 3.1.1

  In terms of enrichment in the aerosol, we observed an increased abundance of Gammaproteobacteria in the chamber aerosol during the *Electra* campaign (see Fig. 5b and S9b). In contrast to the *Electra* campaign, Gammaproteobacteria showed no significant enrichment during the *Oceania* campaign (see Fig. 5a and S9a). An enrichment of Gammaproteobacteria in aerosol has been reported in previous studies in the Baltic Sea (Fahlgren et al., 2010; Seifried et al., 2015; Freitas et al., 2022). However,

in contrast to these studies, we did not find a significant enrichment in Acidimicrobiia, Bacteroidia and Verrucomicrobia and decreased abundances of Actinobacteria and Alphaproteobacteria in SSC compared to the SW during both campaigns. The differences between the previous studies and the current study could be explained by spatial and seasonal differences. Furthermore, both Fahlgren et al. (2010), who conducted measurements at a coastal site in Kalmar and Seifried et al. (2015),





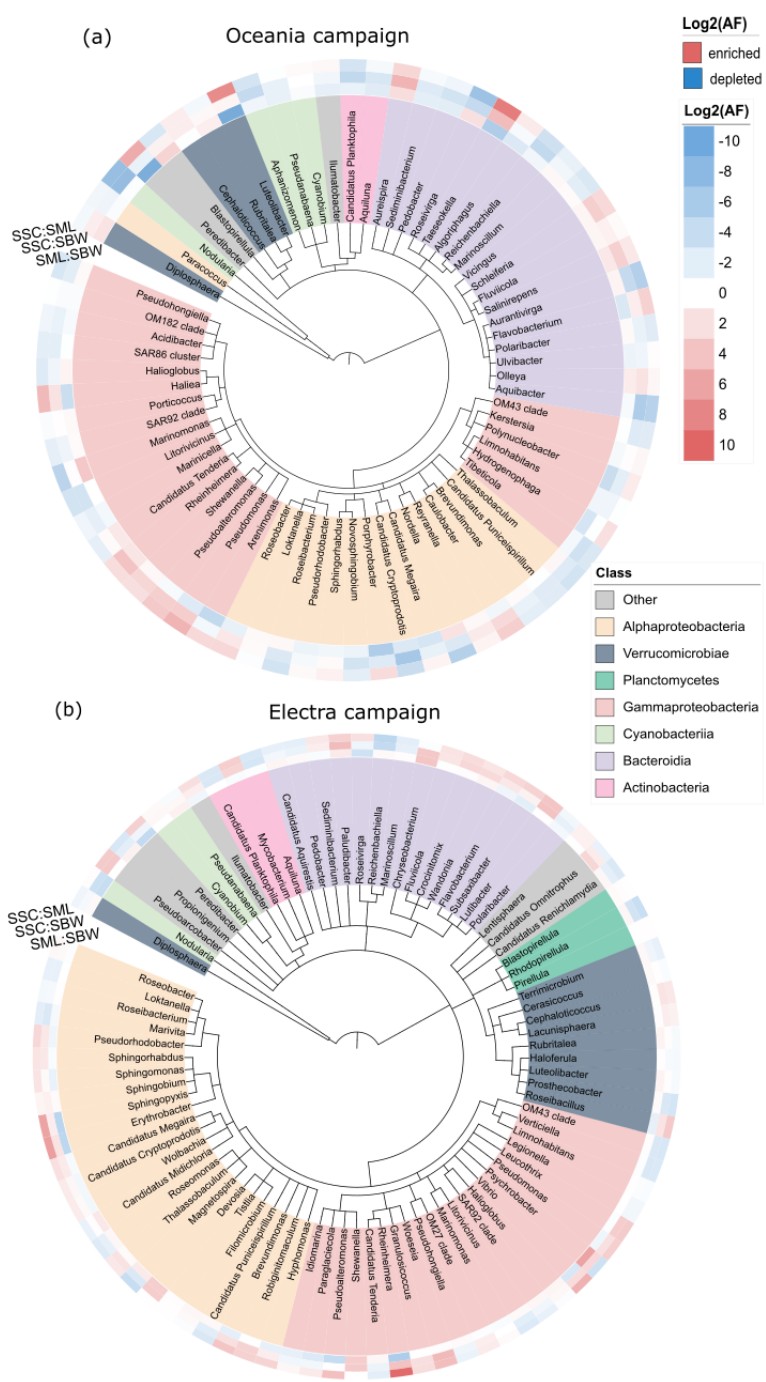

**Figure 4.** Taxonomic trees on genus level for (a) the *Oceania* campaign and (b) the *Electra* campaign of taxa abundant in the bulk seawater (SBW), surface microlayer (SML) and chamber aerosol samples (SSC), color coded according to class. The heat maps indicate enrichment in the SML compared to SBW (inner circle), SSC compared to SBW (middle circle) and SSC compared to SML (outer circle). The tree was generated using PhyloT and iTOL (Letunic and Bork, 2021).





who conducted measurements in the Kattegat, measured in ambient air and as such the community composition from those

studies likely also contained bacteria from terrestrial sources depending on the wind direction.





**Figure 5.** Relative abundances of bacterial classes according to 16S rRNA sequences in sea spray chamber aerosol (SSC), bulk seawater (SW) and surface microlayer (SML) samples collected during the (a) *Oceania* campaign and (b) *Electra* campaign with respective sampling dates.



## 4   Summary and conclusion

We conducted two ship-based campaigns in the Baltic Sea in May and August 2021, where we performed chamber experiments to investigate the emissions of fPBAP with SSA with particular focus on bacteria. By normalizing the bacteria abundances in the aerosol and seawater obtained from fluorescence microscopy with the sodium concentrations in each medium, we found

that bacteria were 13-488 times enriched in the aerosol compared to the bulk seawater. Emission fluxes of bacteria were estimated using two independent approaches: In the first approach, the EFs derived from this study were multiplied with mass emission estimates from existing sea salt parameterizations (Mårtensson et al., 2003; Salter et al., 2015; Zinke et al., 2024c). The second scaling approach is based on a companion study (Zinke et al., 2024c), where we scaled the chamber experiments to EC flux measurements on a nearby island. Using the wind-speed dependent parameterization that was derived from Zinke

et al. (2024c), we derived bacteria and fPBAP emission estimates for a wind speed range between 4-15 m s$^{-1}$. The bacteria emission flux estimates from both approaches agreed fairly well and were on the same order of magnitude as reported values from previous studies. The estimated fPBAP fluxes were however significantly lower. This study is the first to derive bacteria and fPBAP flux estimates from low-salinity (eutrophic) waters such as the Baltic Sea, addressing a research gap outlined by Šantl-Temkiv et al. (2022) and Amato et al. (2023) who emphasized the need for bacteria emission flux estimates over marine

regions.

The 16S rRNA sequencing revealed no significant differences between the bulk seawater and the SML. Similarly, no significant enrichment in terms of cell abundance could be observed in the surface microlayer. One possible explanation for this finding could be that the average wind speed of 6 m s$^{-1}$ encountered during this study generated mixing in the seawater, preventing the enrichment of certain taxa in the SML. Another possible explanation could be dilution of the SML with underlying

bulk water during the glass-plate sampling.

Future studies should conduct long-term ambient measurements to investigate seasonal cycles of PBAP and bacteria emissions covering all seasons and different oceanic waters. These measurements could be incorporated into existing networks, combining meteorological and aerosol measurements to gain a comprehensive understanding of surface-atmosphere exchange processes. Furthermore, future studies should focus on the viability and metabolic activity of airborne microbes, with a partic-

ular focus on the role of phenotypic traits (e.g., pigmentation or hydrophobic surface properties) and atmospheric conditions on their survival and dispersal in the atmosphere. With a potential increase in harmful algae blooms due to climate warming, it is crucial to improve our understanding of whether harmful algae and associated pathogenic microbes can become airborne from seawater and remain viable during aerosolization and atmospheric dispersal. Moreover, determining the fraction of airborne PBAP that can act as ice-nucleating particles is essential for understanding their potential impacts on the climate.

*Data availability.*   The data from this study is available at the Bolin Centre for Climate Research Database (https://doi.org/10.17043/zinke-2024-baltic-bioaerosols-1, Zinke et al., 2024a). Additionally, the sequencing data is publicly available from the NCBI database (accession number PRJNA1110170, http://www.ncbi.nlm.nih.gov/bioproject/1110170).



*Author contributions.* JZ, RF, MS, GF, PZ and EDN designed the experiments. JZ carried out the chamber experiments with help of GF and PZ. The data analysis and visualization was conducted by JZ; GF processed and provided the MBS data. JZ prepared the manuscript with
contributions from all co-authors.

*Competing interests.* At least one of the (co-)authors is a member of the editorial board of Atmospheric Chemistry and Physics.

*Acknowledgements.* We gratefully acknowledge the financial support provided by the Bolin Centre for Climate Research at Stockholm University and the Swedish Research Council (project numbers 2018-05045_VR and 2018-04255_VR). RAF is funded by the Knut and Alice Wallenberg Foundation. The aerosol flux tower and system have been financed initially by FORMAS, project 2007-1362_FORMAS,
and have since received contributions from several FORMAS and VR projects. The ICOS station at Östergarnsholm is maintained by Uppsala University and VR (project 2011-06322_VR). We are obliged to the Institute of Oceanology, at the Polish Academy of Science, who allowed us to use their research vessel, R/V Oceania at no charge. We thank the crew and captain of R/V Oceania and R/V Electra and the technical staff at ACES, Stockholm University, for their support. Thanks goes to Daniel Lundin (Centre for Ecology and Evolution in Microbial model Systems, Linnaeus University, Kalmar, Sweden) for support in the bioinformatic analyses. Additionally, we extend our thanks to Tina Šantl-
Temkiv, Marie Braad Lund and Britta Poulsen from the Department of Biology at Aarhus University, Denmark, for their assistance with sequencing the samples. We are grateful to Christian D.F. Castenschiold (Department of Biology, Aarhus University) for sharing his R-script for generating the scatterplots. Furthermore, thank Pär Helmquist (ACES) for his assistance with the ion chromatography analysis.



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
