# Peer review of "Quantification and characterization of primary biological aerosol particles and bacteria aerosolized from Baltic seawater"

_EGUsphere, 2024_

## Author Comment (AC1)

**Community comment 1**

Hi authors, I have a few questions that I hope the author can help to answer.

Thank you very much for your questions. Below are our responses in blue font.

1.I am wondering why is it possible to substitute the mass fraction of sodium in aerosols for the proportion of sodium in seawater to the total sea salt mass?

Sodium has not been shown to be significantly enriched in sea spray in contrast to calcium and magnesium (Salter et al., 2016), which is also why we used sodium to normalize the bacteria concentrations when calculating the enrichment factors.

2. Figure S4 shows how the authors obtained the mass fractions of these ionic compounds. It is well known that there are a variety of ions in seawater/aerosols, but why do the authors only calculate for these six ionic compounds, so as to overestimate the mass fraction of these ions?

In this study we focused on measuring salt ions in seawater since these were needed to calculate the enrichment factors of bacteria in sea spray aerosols and estimate their fluxes relative to the salt mass. The six ions ($Cl^-$, $Na^+$, $SO_4^{2-}$, $Mg^{2+}$, $Ca^{2+}$, $K^+$) that were measured in the current study account for 99% of the salt mass in seawater.

3. Supplementary Figure S5 appears to be the relative abundance of microorganisms at different sampling sites, rather than the bacterial enrichment factor in SML?

Thanks for pointing this out this error. We have now replaced the figure with the correct figure (now labelled S6).

4. In this paper, the author described two methods to measure bacterial flux, but because the content of organic matter in SSA is unknown, did the author finally use only one method to measure bacterial flux?.

Both methods were used to estimate the bacteria flux and are represented by the blue and green line in Figure 3. We want to clarify that the bacteria flux was not measured but estimated using two different approaches: 1) by multiplying the enrichment factor derived from this study with the mass flux based on a previous parameterization derived from a companion study (represented by the green line) and 2) by multiplying the ratio of bacteria to aerosols with the number flux from the same companion study (represented by the blue line). Accurate estimation of SSA density, necessary for approach 1, would benefit from knowledge of the SSA's organic content, but measuring this was beyond our study's scope. Using a density for higher organic content would only slightly shift the estimated bacteria flux.

**References**

Salter, M.E., Hamacher-Barth, E., Leck, C., Werner, J., Johnson, C.M., Riipinen, I., Nilsson, E.D. and Zieger, P., 2016. Calcium enrichment in sea spray aerosol particles. *Geophysical Research Letters*, *43*(15), pp.8277-8285.

---

## Author Comment (AC2)

We thank the reviewers for their useful comments. Below is a detailed, point-by-point list of our replies to the comments of the reviewers. Our responses are in blue font and the reviewers' comments are in black font.

**Reviewer 1**

The presented study conducted two ship-based campaigns in the Baltic Sea in May and August 2021, and performed chamber experiments to investigate the emissions of biological particles with SSA with particular focus on bacteria. The experimental design was very comprehensive, and the results are useful to understand the air-sea exchange of bacteria. The manuscript is presented well, and I have only several comments to be clarified, mainly related to the methodology and the explanation of some results.

General comments:

DAPI staining also dyes fungal spores and other biological particles containing DNA blue under the UV excitation. How did the authors distinguish bacterial cells from other biological particles for FM enumeration? How large were the measured bacterial cells? The authors give some two reasons (Line 261-267) why the concentration of airborne bacteria 30-40 times higher than that of fPBAP (Figure 2, Line 249-257). For the first reason, were viruses and cell fragments enumerated using FM? For the second reason, did the author find some gels in FM images (e.g., Fig. 4 in Aller et al. 2005)? Because sonication was only 1 minute, some gels may still exist.

The reviewer rightly pointed out that DAPI also stains other biological particles containing DNA such as fungal spores. Since we cannot entirely exclude the possibility that fungal spores and other biological material might have been included in our DAPI-stained cell counts, we have now replaced "bacteria" with "microbes" or "microbial cells" throughout the manuscript and added the following text to section 2.2: "Positively stained cells **(including bacteria, fragments, spores and large viruses)** were counted […]".
Indeed, some gels still existed after sonication. However, these were excluded from analysis. We counted a cell only if it displayed a well-defined cell shape with distinct edges, and if its fluorescence was significantly higher than the background fluorescence, which could be attributed to organics. That being said, cell fragments and viruses were not explicitly excluded from analysis and could have caused an overestimation of cells using fluorescence microscopy (FM). In Zinke et al. (2024b), where we described the results from similar experiments conducted in the North-Eastern Atlantic Ocean, we discussed this issue in more detail. We added boxplots of the cell sizes from FM and fPBAP sizes from the online measurements measured during the current study to the supplement (see Figure below). 99% of cells from FM were smaller than 2.8 µm, while 39% of fPBAP were larger than 2.8 µm.
Another factor that could lead to an overestimation of cells using FM is the sonication step that likely caused a break-up of cell clusters which would only be enumerated as one bigger fPBAP particle using the MBS.

[Figure]

*Figure S7: Boxplots of cell diameters obtained from fluorescence microscopy (FM) and optical diameters of primary biological aerosol particles (PBAP) obtained from the multiparameter bioaerosol spectrometer (MBS) measurements.*

Figure 2: Why was the concentration of fPBAP likely quite stable? It is almost unchanged, especially when the chamber was operated in closed mode. What is the possible reason? Could it be a background signal? Are there any variations in the concentrations of CP and FP? Why not illustrate them in the figure?

Below is a figure showing the time series of the fPBAP/CP ratio, with periods of closed-mode chamber operation highlighted. The figure shows that the ratio was higher at lower jet speeds and decreased when the jet flowrate increased. Focusing on the higher flowrate periods, the median values for flow-through and closed mode were nearly identical (0.000737 and 0.0007327, respectively). However, a ~17% decrease in the ratio was observed on the nights of August 18 and 22. In other closed-mode periods without a decrease, removal by aerosolization and cell multiplication likely balanced each other, suggesting that the chamber's large volume provided a sufficient reservoir of fPBAP.
We added the following sentence to the manuscript: "Notably, fPBAP represented only a minor fraction of all (median 1.1%) and an even smaller proportion of CP, which decreased from 0.09% to 0.07% when the jet flow rate was increased but remained fairly constant even when the chamber was operated in closed mode (see fig. S4). This suggests that the chamber's large volume provided a sufficient reservoir of fPBAP to balance cell multiplication with the removal by aerosolization."

[Figure]

*Figure S4: Time series of the ratio of fPBAP to coarse particles (CP). Times when the chamber was operated in flow-through mode are marked in blue and closed-mode periods are marked in orange.*

We have now added CP and FP to the figure 2 in the manuscript. Neither of them show strong variations. As such, it seems that the aerosolization rate was stable over the duration of the campaign. Factors that could affect the aerosolization rate are changes in seawater properties such as seawater temperature, salinity and biological activity. However, from panels b) and c) it appears that none of these factors changed drastically during the campaign. The reviewer also mentioned the closed-mode operation in particular. While the seawater temperature changed slightly during these time periods, this increase was still small enough to not affect the SSA production rate (stronger changes in SSA production with seawater temperature would be expected between temperatures of 0-10℃).

Line 247: "This contrasts with previous studies that have reported higher bacterial concentrations in the SML". Why was the result contrast to previous studies, e.g., Aller et al., 2005? The authors give two explanations in Line 282: What was the range of wind speed? Can SML form or keep at those wind speeds? How much was the thickness of sampled SML? Line 277-279, "This decrease could be attributed to the higher jet flow rate potentially preventing the formation of a SML, which typically contains higher concentrations of bacteria and organic material." This explanation should also be based on the fact that bacteria were enriched during transport from subsurface waters to the SML. If there was no enrichment, the explanation is false.

The wind speed during the Electra campaign ranged between 0 and 14 m s$^{-1}$. Below we have added a histogram of the wind speeds encountered during the Electra campaign that is published in the supplementary Figure S5a in Zinke et al. 2024c. As can be seen from this figure, most wind speeds were greater than 4 m s$^{-1}$ at which wind-induced wave breaking starts to occur. While the surface microlayer can still persist at wind speeds above 4 m s$^{-1}$, the turbulent mixing at wind speeds > 4 m s$^{-1}$ inhibits the enrichment of bacteria in the SML (Rahlff et al., 2017). We have reformulated the sentence pointed out by the reviewer as follows: "This decrease could be attributed to the higher jet flow rate potentially preventing the **microbial enrichment in the SML"**.

[Figure]

*Figure S5a (from Zinke et al. 2024c): Frequency histogram of wind speeds encountered during the Electra campaign.*

During the Electra campaign a glass plate was used to sample the SML. Unfortunately, we did not record the number of dips used to sample the SML, which would be necessary to estimate the thickness of the sampled SML. As a rough estimate glass plate samplers typically have sampling depths of 20–150 µm while mesh screen samplers (as was used during the Oceania campaign) typically sample the uppermost 150–400 µm (Cunliffe & Wurl, 2017). According to Zhang et al. (1998) the SML is typically 50 µm +/- 10 µm deep. As such, both methods will collect a diluted SML sample (but more so for the mesh sampler). We have added the following sentences to the text in section 2.1: "Glass plate samplers typically have sampling depths of 20–150 µm while mesh screen samplers have

sampling depths of 150–400 µm (Cunliffe & Wurl, 2017)." The discussion of the dilution of the SML sample during sample collection is already addressed in the discussion.

Line 258-260: Did the authors measure the abundance of biological and fluorescent matter or any other proxies, for instances, Chl a, in surface water during August 2021 and June 2018? If measured, it is better to note them directly.

Chlorophyll-a concentrations in this study were estimated using reanalysis data from satellite measurements and showed high concentrations of 5.2±1.1 mg m$^{-3}$ in August 2021. In June 2018, chlorophyll-a concentrations were measured with a fluorometer inside an onboard monitoring box and were generally lower than 2 mg m$^{-3}$. We have added this information to the text: "(i.e. chlorophyll-a was < 2 mg m$^{-3}$ in June 2018 compared to 5.2 ± 1.1 mg m$^{-3}$ in August 2021)"

Line 366: Why did the author use "a factor of 0.00086 (mean, range 0.0003 - 0.0013)"? The authors mentioned "Notably, fPBAP represented only a minor fraction of all FP (1.4±2.7%) and an even smaller proportion of CP (0.15±0.38%)". Was the fraction of bacteria cells determined from fluorescence microscopy to the total aerosol concentration larger than 0.15% in this study? Again, the measured size ranges of bacterial cells and total aerosol particles should be the same to compare.

The ratio of 0.15 mentioned by the reviewer is the mean ratio of fPBAP to CP (with diameters larger than 0.8 µm). We now present this value as median values (see reply above). In contrast, the factor of 0.00086 is the ratio of bacteria to the total aerosols in the size range 0.02 < Dp < 2.8 µm. This size range was chosen for the sake of comparability since the Mårtensson parameterization is limited to this size range. We have added the following sentences to the text: "While the majority of microbial cells (99%) that were measured with FM were smaller than 2.8 µm (see Figure S7), 39% of fPBAP were larger than 2.8 µm. As such, using the same range to estimate fPBAP fluxes would result in a grave underestimation of the fPBAP flux. Instead we calculated the ratio of fPBAP to total aerosols for the entire size range measured by the DMPS and OPSS (0.015 < Dp < 10 µm) which was 2.209 · 10$^{-5}$ (mean, range 9.945·10$^{-6}$ – 4.627·10$^{-5}$). The fPBAP emission flux was then estimated by multiplying this factor with the total aerosol number flux integrated over this size range"

Sect. 3.1.4: As mentioned in the abstract "16S rRNA sequencing identified the diversity of bacteria enriched in the nascent SSA compared to the underlying seawater." But the description in this section does not mention it clearly, and the authors mentioned the difference/enrichment in the R/V Oceania campaign was not obvious. There are no statistic results for these comparisons, for instance, the values of richness index. Was the difference in the R/V Electra campaign significant?

The difference between aerosol and seawater/SML samples was found to be significant using the ANOSIM statistical test as described in lines 401-404. The richness index for chamber aerosol, SML and seawater samples collected during both campaigns is presented in Figure S9a (previously S8a) in the supplement. We have added the mean values and standard deviations in SSC, SW and SML from both campaigns to the text:" Regarding alpha diversity richness index, the samples from the R/V Oceania campaign showed slightly less diversity **(270± 54 in SSC, 315 ± 69 in SW, 313 ± 32 in SML)** than those from the R/V Electra campaign **(427 ± 58 in SSC, 472 ± 38 in SW, 379±61 in SML**, see also Fig. S9a)"

 We further added the relative abundances (in percent) of different bacteria classes in chamber aerosols compared to their relative abundance in seawater to section 3.1.4. The discussion of the airborne microbial community now reads as follows: "In terms of enrichment in the aerosol, we observed an increased abundance of Gammaproteobacteria in the chamber aerosol with relative

abundances of 23% compared to 12% in the seawater during the Electra campaign (see Fig. 5b). In contrast to the Electra campaign, Gammaproteobacteria showed no significant enrichment during the Oceania campaign with relative abundances between 17-19% in both chamber air and seawater (see Fig. 5a). An enrichment of Gammaproteobacteria in aerosol has been reported in previous studies in the Baltic Sea (Fahlgren et al., 2010; Seifried et al., 2015; Freitas et al., 2022). However, in contrast to these studies, we did not find a significant enrichment in Acidimicrobiia, Bacteroidia and Verrucomicrobia during the Electra campaign (with relative abundances of 10-11% in all sample types) and a depletion of Bactereroidia during the Oceania campaign (14% in the chamber air compared to 21% in seawater). Furthermore, we observed decreased abundances of Actinobacteria and Alphaproteobacteria in chamber air compared to the seawater during both campaigns (2% Actinobacteria and 9-10% Alphaproteobacteria in chamber air compared to 11% Actinobacteria and 24-25% Alphaproteobacteria in seawater during both campaigns). The differences between the previous studies and the current study could be explained by spatial and seasonal differences. Furthermore, both Fahlgren et al. (2010), who conducted measurements at a coastal site in Kalmar and Seifried et al. (2015), who conducted measurements in the Kattegat, measured in ambient air and as such the community composition from those studies likely also contained bacteria from terrestrial sources depending on the wind direction. During the *Electra* and *Oceania* campaign we observed an enrichment of Cyanobacteria in the SSC (with relative abundances in SSC of 27% and 52% compared to 12% and 28% in SW, respectively). Similarly, Lewandowska et al. (2017), who conducted ambient measurements of bioaerosols over land (Gdynia) and at sea (Gulf of Gdansk, Southwestern Baltic), identified picoplanktonic Cyanobacteria in the air."

Technical comments:

Line 19: cloud and ice condensation nuclei---> cloud condensation nuclei and ice nuclei

We have made the changes as suggested by the reviewer.

Table S1 and S2: what do the blanks, (X) and X mean? Does (X) mean that the analysis was not conducted?

As specified in the table caption, samples marked with (X) were excluded from analysis (e.g. due to an issue with the DAPI staining during the Oceania campaign). The collection of filter blanks is described in section 2.1: These were collected to determine the background contamination by placing the filters in their respective filter holders without drawing any air through them and immediately removing them again. During the Oceania campaign we also collected a sample of the ultrapure water (MiliQ) to determine the background concentration for the ion chromatography method.

S4: The genus and species should be in italic, and the name of each species should consist of two parts: the genus name and the specific epithet (species name).

We made the changes as suggested by the reviewer.

Line 123: Bacterial enumeration

We have changed "bacteria enumeration" to "bacterial enumeration" here and throughout the text.

Line 194: WELAS--->OPSS

We have changed "WELAS" to "OPSS" as rightly pointed out by the reviewer.

Line 199: MBS detects and sizes particles?

We have rephrased the sentence as follows: "The MBS uses a low-power laser with a wavelength of 635 nm to detect the incoming particles and the scattering signal to determine the size of each individual particles."

Line 231: 5.2±1.1 mg m−3

We have removed the typo.

Line 413: add "richness index"?

We have added "richness index" after "alpha diversity" as suggested by the reviewer.

**Reviewer 2**

Authors presented a study based on two campaigns performed on ships in Baltic Sea with aim to expand the knowledge on primary biological particle (PBAP) aerosolization with sea spray aerosols (SSAs). To achieve this goal, aerosols were generated in highly controlled plunging jet sea spray simulation chamber that circulates bulk sea water with ship's flow-through system at 1.5m depth. The aerosol size distribution was measured with DMPS and OPSS, and fluorescent particles were measured online with an MBS. In addition to online measurements, aerosols were collected on filters for following bacterial enumeration with fluorescence microscopy, bacterial specie composition determination with 16S rRNA metabarcoding and ion chromatography, and sea water samples were collected at filter exchanges. The collected data was used to estimate bacterial emission fluxes, enrichment of bacteria in SSA compared to bulk sea water and further, enrichment of specific classes in SSA compared to sea water.

**General comments**

- There have been very few studies attempting to estimate the emission flux of bacteria in SSA and to determine the enrichment factors for different species to date. Authors presented estimations using two approaches and compared to the previously published estimations. However, the estimations rely heavily on parametrizations derived by Mårtensson et al (2003) restricting application of these estimations to the size range of the $0.02 < D_p < 2.8$ μm.
- Authors encountered two controversial findings during the study: no difference in specie composition between sea microlayer samples and bulk sea water, and occurrence of ASVs in aerosol samples not observed in bulk sea water that these species were aerosolized from. These observations were discussed in the manuscript to an extent, but would require further clarifications (see specific comments).
- Figures, and supplementary material captions to the figures and tables require refinement. More details in specific and technical comments.

**Specific comments**

Lines 38-39: "Recent observations indicate that the contribution of the SML to SSA is relatively small compared to bubbles originating from subsurface water (Chingin et al., 2018; Frossard et al., 2019)".

Do the authors mean the contribution of SML to production of SSA from jet drops? Or from the point of view of only biological SSA, which are mainly concentrated in jet drops? As is, doesn't this statement contradict with the findings of Wang et. al (2017) you mentioned in previous sentence, who indicated that the fraction of SSA produced with jet drops varies from 20% in natural sea water to 43% during algal bloom?

The studies by Chingin et al. and Frossard et al. concern the enrichment of surfactants on the bubble surface while they are rising to the surface versus enrichment from the surface microlayer. We removed this sentence given that our study did not investigate surfactant enrichment.

Lines 118-119: "However, due to poor staining, bacteria could not be confidently enumerated during the R/V Oceania campaign, so these samples will only be discussed in terms of bacterial community composition." Were there any differences in sample handling that led to poor staining? Was the same staining method used for both campaigns or was it improved for the *Electra* campaign?

During the *Oceania* campaign, filter samples were stained directly on the filter. However, using this approach we were not able to remove the salts that accumulated on the filters during sampling. The salt accumulation resulted in a high autofluorescence/background that interfered with the fluorescence of the microbial cells. To avoid this issue, we adapted the staining approach during the *Electra* campaign as described in section 2.2, where we extracted the filter samples in 5 mL ultrapure water, fixed, stained and filtered the suspension prior to enumeration. We also fixed, stained and enumerated the extracted filter to estimate the extraction efficiency of the sonication step and to account for cells that may have remained on the extracted filter. All values presented in the manuscript account for the extraction efficiency. We have added the following sentence to section 2.2: "To assess the extraction efficiency and account for cells that may have remained on the extracted filter, these were also fixed, stained and analyzed alongside the extracts." To clarify that this approach was only used during the *Electra* campaign we added "During the *Electra* campaign" at the beginning of the paragraph.

Section 2.2: Bacterial enumeration was performed using DAPI staining of particles larger than 0.2 µm in diameter and fluorescence microscopy. DAPI stains dsDNA in all PBAPs, did you differentiate bacteria from other species, for example fungi or diatoms, based on morphology of the observed cells? If yes, what were the criteria for classification of the cells as bacterial? What was the size range of the cells?

The reviewer rightly pointed out that DAPI also stains other biological particles containing DNA such as fungal spores. We have addressed this comment in detail in the reply to reviewer 1. In summary, we counted a cell only if it displayed a well-defined cell shape with distinct edges, and if its fluorescence was significantly higher than the background fluorescence. However, we did not distinguish between bacteria and other biological material and therefore we have replaced "bacteria" with "microbes" or "microbial cells" throughout the manuscript and added the following text to section 2.2: "Positively stained cells **(including bacteria, fragments, spores and large viruses)** were counted […]"

We added a boxplot of cells sizes from FM and fPBAP sizes from MBS measurements to the supplement (see Figure below). 99% of cells from FM were smaller than 2.8 µm, while 39% of fPBAP were larger than 2.8 µm.

[Figure]

*Figure S7: Boxplot of cell diameters obtained from fluorescence microscopy (FM) and optical diameters of primary biological aerosol particles (PBAP) obtained from the multiparameter bioaerosol spectrometer (MBS) measurements.*

Figure 2: The bacterial concentration determined with fluorescence microscopy for dates 16/08-17/08 and 18/08 seem to have a slight decline though according to Figure, the chamber was in closed state. Is this observation significant in terms of sensitivity of the analysis? Would you contribute this decline in bacterial concentration to the cell disruption with plunging jet? If so, has the system been tested for disruption of microorganisms in closed state for prolonged periods of time?

During a similar study, where we conducted closed mode mesocosm experiments in the Azores archipelago (Zinke et al., 2024b), we found that the closed mode operation led to a depletion of certain taxa in the seawater in the mesocosm over a time period of 48 hours. As such, running the chamber in closed mode might indeed affect the number of bacteria that are aerosolized. In the reply to reviewer 1, we showed that there was a slight decrease in fPBAP in comparison to CP on the nights of August 18 and 22. In other closed-mode periods no such decrease was observed, suggesting that the removal by aerosolization and cell multiplication likely balanced each other and that the chamber's large volume provided a sufficient reservoir of fPBAP. It should be noted that the chamber used in the current study was ten times larger than the one used in the Azores study and closed-mode periods were shorter than during the Azores study.

We also want to add that on the dates mentioned the water was replaced on the 17/08 noon during transit from Fårösund to Herrvik and thereafter the chamber was operated in closed mode again. As such, the samples collected during the 16-17/08 and 18/08 are independent samples.

Section 3.1.1: According to your results, sea microlayer (SML) samples did not differ from bulk sea water (SW) samples in bacterial species composition, during both ship campaigns. However, the R/V *Electra* campaign coincided with the bloom of filamentous cyanobacteria described in the results 3.1. and shown with calculated chlorophyll a concentration. This should show in high abundancy of 16S rRNA of cyanobacteria in the SML samples compared to SW samples, which was not the case in this study. This finding was explained by high windspeed and potential mixing of SML with underlying sea water during sampling. Unfortunately, the Figure S5 in Supplements is a copy of Figure 5, which does not indicate the relation of enrichment factor compared to the wind speed. Could the figure be updated? The wind speed was high during *Electra* campaign, was it also the case for *Oceania* campaign? If the mixing of SML with underlying water was due to sampling method, could you evaluate the reliability the results from *Oceania* campaign SML samples? Could the vicinity

of ship affect the collection SML samples? In case if high windspeeds, could movement of the ship lead to mechanical mixing of surrounding waters?

We have now updated Figure S5 with the correct figure (now labelled S6):

[Figure]

*Figure S6 Microbial enrichment factor in the SML compared to underlying seawater versus wind speed during the Electra campaign.*

The wind speed during the *Oceania* campaign was comparable to the wind speed during the *Electra* campaign (see histograms below, taken from the supplement of Zinke et al., 2024c). During the *Oceania* campaign, the SML was sampled using a mesh screen that was lowered from the side of the ship, while during the *Electra* campaign the SML was sampled using the glass plate method from a dinghy with some distance from the ship. The difference in sampling approach might have impacted the SML thickness that was sampled. As discussed in reply to reviewer 1, the mesh screen sampler has a sampling depth of 150-400 µm (Cunliffe & Wurl, 2017), while glass plate samplers typically have sampling depths of 20–150 µm. As such the SML samples collected with the mesh screen were likely more diluted than those collected with the glass plate method. Furthermore, we cannot exclude the possibility that the proximity to the ship affected the collection of the SML during the *Oceania* campaign.

[Figure]

*Figure: Frequency histograms of wind speeds encountered during the (a) Electra campaign and (b) Oceania campaign (from Zinke et al., 2024c).*

Section 3.1.3: The bacterial emission fluxes were estimated using two approaches, both for bacteria in size range of 0.02 < $D_p$ < 2.8 µm to which Mårtensson et al (2003) parametrization is valid, whereas the bacteria enumerated in campaign was sized larger than 0.2µm. Was it was accounted in the emission fluxes calculated here? What would be the estimate of the significance of emission of the bacteria and bacterial agglomerates larger than 2.8µm?

As can be seen from the boxplots (Figure S7), the majority of microbial cells were smaller than 2.8 µm, as such the emission flux of cells would not be significantly different (<1% difference) if we had used the entire size range of aerosol particles (0.015 < Dp < 10 µm). However, 39% of fPBAP were larger than 2.8 µm and if we had used the smaller size range (0.02<Dp<2.8 µm) for fPBAP fluxes, we would have gravely underestimated the fPBAP flux. As such we had initially calculated these fluxes for the larger size range (0.015 < Dp < 10 µm). We realized that this information was missing from the text and added the following sentences:

"While the majority of microbial cells (99%) that were measured with FM were smaller than 2.8 µm (see Figure S7), 39% of fPBAP were larger than 2.8 µm. As such, using the same range to estimate fPBAP fluxes would result in a grave underestimation of the fPBAP flux. Instead, we calculated the ratio of fPBAP to total aerosols for the entire size range measured by the DMPS and OPSS (0.015<Dp<10 µm) which was $2.209 \cdot 10^{-5}$ (mean, range $9.945 \cdot 10^{-6}$-$4.627 \cdot 10^{-5}$). The fPBAP emission flux was then estimated by multiplying this factor with the total aerosol number flux integrated over this size range."

In the section 3.1.4, among the taxa observed in air samples, there were unique ASVs not found among SW and SML. Authors state that "One possible explanation for this could be selective aerosolization of certain taxa.". If the selective aerosolization occurred, shouldn't the same ASVs be also observed in SW and SML samples, but in lower abundancies? What is the limit of detection with chosen extraction and amplification method used? Authors suggest that that potentially some taxa may be "entirely removed from the seawater sample" due to selective aerosolization, but there are 357 ASVs from Oceania campaign and 102 ASVs from Electra campaign. How realistic is this this possibility? Another potential explanation was "the presence of free DNA in the head space of the chamber that can pass through the HEPA-filters". HEPA filters are 99.97% efficient in filtering air and given intensive SSA production with plunging jet, shouldn't the possible contamination through HEPA filters be much smaller than the observed number of unique ASVs in SSC samples? What were the controls and were there any indication of contaminants in them? Could this be results of over-amplification due to triple PCR? Could it be misclassification introduced by bioinformatic pipeline?

The reviewer raises an important issue here. We realize that there might have been some confusion due to the poor choice of the word "unique" in this context. These numbers are actually based on the Venn diagram (now Figure S8), which compares ASVs in seawater and the chamber air from both campaigns. To avoid any confusion we rephrased the sentence as follows: "We compared the ASVs present in SW and SSC samples from both research campaigns. Our analysis revealed that 3033 ASVs were common to all samples, regardless of type or campaign. In SW samples, 773 ASVs **were solely detected during** the R/V *Electra* campaign and 366 ASVs **were solely detected during** the R/V *Oceania* campaign. For SSC samples, 357 ASVs were **only detected during** the R/V *Oceania* campaign and 102 ASVs **were only detected during** R/V *Electra* campaign."

That being said, the number of ASVs detected in SSC but not in SW only constitute 2-3% for both campaigns. It should also be considered that the majority of aerosols produced in the chamber was captured by the filters, but only 500 mL from a total of 100 L of seawater in the tank was sampled for DNA extraction. As such it might be possible that certain rare ASVs were not present in this subsample of seawater.

The reviewer mentioned the efficiency of the HEPA filter. Such filters capture particles that are 0.3 µm or larger with an efficiency of 99.97%. To ensure the chamber air was particle-free, we also measured the particle concentration prior to the experiments with the plunging jet turned off using a CPC with a size threshold of 10 nm and waited until the concentration reached 0 cm$^{-3}$. However, since free DNA molecules can be smaller than the CPC cut-off (in the range of 1-2 nm) we cannot completely dismiss the possibility that small, free-floating DNA molecules could have passed through the filter. Furthermore, there is a chance that these free-floating DNA fragments might have been subsequently over-amplified by the PCR.

One approach to determine the detection limit is to collect control samples and remove the ASVs occurring in these controls from further analysis. Using the R decontam package, 175 contaminant ASVs were identified in handling blank samples and removed from further analysis in this study (see also Table S3 for a list of these samples).

Concerning the possibility of misclassifications by the pipeline: The nf-core ampliseq pipeline that was used to assign the taxonomy gives a bootstrap value for each rank and chooses the most specific rank with a bootstrap value larger than 0.5. However, we cannot not fully exclude the possibility that misclassification was introduced by the pipeline due to limitations of the SILVA reference database, sequencing artifacts or chimeras that might have slipped through the correction steps in the pipeline.

We have added the following sentence to the text: "Subsequently, these DNA fragments might have been overamplified by the triple-PCR. Furthermore, we cannot not fully exclude the possibility that misclassification were introduced by the pipeline due to limitations of the reference database, sequencing artifacts or chimeras that might have slipped through the correction steps in the pipeline ."

Lines 426-435: The enrichment of species in SSC samples compared to SW and SML requires more detailed look. In lines 429-431 "However, in contrast to these studies, we did not find a significant enrichment in Acidimicrobiia, Bacteroidia and Verrucomicrobia and decreased abundances of Actinobacteria and Alphaproteobacteria in SSC compared to the SW during both campaigns." However, the Figure 5b indicates clear decrease in *Actinobacteria* and *Alphaproteobacteria* in SSC samples compared SW and SML samples, while the abundance changes in *Gammaproteobacteria* described in lines 426-428 is much more subtle. Was there statistical analysis performed to support the observations? Could you also discuss the enrichment of *Cyanobacteria* in SSC compared to SW and SML?

We have now added the relative abundances in SSC, SW and SML to the discussion in section 3.1.4. The paragraph now reads as follows: "In terms of enrichment in the aerosol, we observed an increased abundance of Gammaproteobacteria in the chamber aerosol with relative abundances of 23% compared to 12% in the seawater during the Electra campaign (see Fig. 5b). In contrast to the Electra campaign, Gammaproteobacteria showed no significant enrichment during the Oceania campaign with relative abundances between 17-19% in both chamber air and seawater (see Fig. 5a). An enrichment of Gammaproteobacteria in aerosol has been reported in previous studies in the Baltic Sea (Fahlgren et al., 2010; Seifried et al., 2015; Freitas et al., 2022). However, in contrast to these studies, we did not find a significant enrichment in Acidimicrobiia, Bacteroidia and Verrucomicrobia during the Electra campaign (with relative abundances of 10-11% in all sample types) and a depletion of Bactereroidia during the Oceania campaign (14% in the chamber air compared to 21% in seawater). Furthermore, we observed decreased abundances of Actinobacteria and Alphaproteobacteria in chamber air compared to the seawater during both campaigns (2% Actinobacteria and 9-10% Alphaproteobacteria in chamber air compared to 11% Actinobacteria and 24-25% Alphaproteobacteria in seawater during both campaigns). The differences between the previous

studies and the current study could be explained by spatial and seasonal differences. Furthermore, both Fahlgren et al. (2010), who conducted measurements at a coastal site in Kalmar and Seifried et al. (2015), who conducted measurements in the Kattegat, measured in ambient air and as such the community composition from those studies likely also contained bacteria from terrestrial sources depending on the wind direction. During the *Electra* and *Oceania* campaign we observed an enrichment of Cyanobacteria in the SSC (with relative abundances in SSC of 27% and 52% compared to 12% and 28% in SW, respectively). Similarly, Lewandowska et al. (2017), who conducted ambient measurements of bioaerosols over land (Gdynia) and at sea (Gulf of Gdansk, Southwestern Baltic), identified picoplanktonic Cyanobacteria in the air."

Figure 5: The results for different sample types from the same days are difficult to compare in the given format. Arranging the X-axis temporally would help comparing the results. Also, in panel (b), sample SW_210811 is presented twice, with different data. Which one is the correct one?

These are two different samples, one was collected at 06:00 local time (LT) while the other was collected at 14:00 LT (see also table S2). We added the times to the dates and re-arranged the x-axis temporally as suggested by the reviewer.

Figure S4: The x-axes are undescriptive; I would suggest changing them to be by date and below describes the sample type – bulk sea water and SML samples separately.

We have now changed the x-axes label as per suggestion of the reviewer. The figure is now labelled S5.

**Technical corrections**

Lines 29-32: Please add references to the text.

We have added the following references to the text: Després et al., 2012; Blanchard, 1963 and 1983; Lewis and Schwartz, 2004.

Line 31: Words "film drop" is written twice in a row, remove one repetition.

We have removed the repetition of "film drops".

Lines 48-49: "Recent advancements in real-time single-particle analysis instruments using ultra-violet light-induced fluorescence allow continuous monitoring of fluorescent (fPBAP)". I suggest changing "… fluorescent (fPBAP)" to "… fluorescent PBAP (fPBAP)" and to add few examples of aforementioned instruments.

We have now added "PBAP" as well as the following examples of bioaerosol instruments: "Examples for real-time single-particle analysis instruments are the Wideband Integrated Bioaerosol Sensor (WIBS), the Multiparameter Bioaerosol Sensor (MBS, Ruske et al., 2017}, the Spectral Intensity Bioaerosol Spectrometer (SIBS, Könemann et al., 2019), the PA-300 (Kiselev et al., 2013) and the Rapid-E (Sauliene et al., 2019)."

Figure 1: "Dry air genator" -> "Dry air generator". In caption "mulitparameter" -> "multiparameter"

We have corrected these typos.

Figure 4: Could the same classes be shown in one segment? For example, *Gammaproteobacteria* class is shown in two segments in panel (a).

The taxonomic trees were generated using iTOL. Unfortunately, we did not find a way to modify the appearance of the tree from the Oceania campaign to completely avoid the splitting of all class segments.

Table S1: Insert description of the different abbreviations used in the table to the caption.

We have added descriptions of the abbreviations to the table caption for table S1 and S2.

Figure S5: same as Figure 5, repeated by accident? Update the figure to match the caption.

We have now updated the figure with the correct figure (now labelled S6).

**References**

Cunliffe, M. and Wurl, O.: Sampling the sea surface microlayer. Hydrocarbon and lipid microbiology protocols: field studies, pp.255-261, 2017

Blanchard, D. C.: The electrification of the atmosphere by particles from bubbles in the sea, Prog. Oceanogr., 1, 73–112, https://doi.org/10.1016/0079-6611(63)90004-1, 1963.

Blanchard, D. C.: Air-Sea Exchange of Gases and Particles, chap. The Production, Distribution, and Bacterial Enrichment of the Sea-Salt Aerosol, pp. 407–454, D. Reidel Publishing Company, 1983.

Després, V., Huffman, J. A., Burrows, S. M., Hoose, C., Safatov, A., Buryak, G., Fröhlich-Nowoisky, J., Elbert, W., Andreae, M., Pöschl, U., et al.: Primary biological aerosol particles in the atmosphere: a review, Tellus B: Chemical and Physical Meteorology, 64, 15 598, 2012.

Könemann, T., Savage, N., Klimach, T., Walter, D., Fröhlich-Nowoisky, J., Su, H., Pöschl, U., Huffman, J. A., and Pöhlker, C.: Spectral Intensity Bioaerosol Sensor (SIBS): an instrument for spectrally resolved fluorescence detection of single particles in real time, Atmospheric Measurement Techniques, 12, 1337–1363, 2019.

Konik, M., Kowalewski, M., Bradtke, K., and Darecki, M.: The operational method of filling information gaps in satellite imagery using numerical models, International Journal of Applied Earth Observation and Geoinformation, 75, 68–82, 2019.

Lewis, E. R. and Schwartz, S. E.: Sea Salt Aerosol Production: Mechanisms, Methods, Measurements and Models - A Critical Review, Geophysical Monograph Series, Vol. 152, American Geophysical Union„ 2004

Ruske, S., Topping, D. O., Foot, V. E., Kaye, P. H., Stanley, W. R., Crawford, I., Morse, A. P., and Gallagher, M. W.: Evaluation of machine learning algorithms for classification of primary biological aerosol using a new UV-LIF spectrometer, Atmospheric Measurement Techniques, 10, 695–708, 2017

Šauliene, I., Šukiene, L., Daunys, G., Valiulis, G., Vaitkevicius, L., Matavulj, P., Brdar, S., Panic, M., Sikoparija, B., Clot, B., et al.: Automatic pollen recognition with the Rapid-E particle counter: the first-level procedure, experience and next steps, Atmospheric Measurement Techniques, 12, 3435–3452, 2019

Zhang ZB, Liu LS, Wu ZJ, Li J, Ding HB: Physicochemical studies of the sea surface microlayer - I. Thickness of the sea surface microlayer and its experimental determination. J Colloid Interface Sci 204:294–299, 1998

Zinke, J., Freitas, G., Salter, M. E., Lundin, D., Aggarwal, S., Zieger, P., Mohr, C., and Foster, R. A.: Quantification and characterization of bacteria emission over the North-Eastern Atlantic using mesocosm experiments, Environmental Science and Technology Air, https://doi.org/10.1021/acsestair.3c00017, 2024b.

Zinke, J., Nilsson, E. D., Markuszewski, P., Zieger, P., Mårtensson, E. M., Rutgersson, A., Nilsson, E., and Salter, M. E.: Sea spray emissions from the Baltic Sea: comparison of aerosol eddy covariance fluxes and chamber-simulated sea spray emissions, Atmospheric Chemistry and Physics, 24, 1895–1918, https://doi.org/10.5194/acp-24-1895-2024, 2024c.